# Paradoxical neuronal hyperexcitability in a mouse model of mitochondrial pyruvate import deficiency

Andres De La Rossa[1†], Marine H Laporte[1*†], Simone Astori[2†], Thomas Marissal[3,4], Sylvie Montessuit[1], Preethi Sheshadri[5], Eva Ramos-Fernández[2], Pablo Mendez[6], Abbas Khani[4], Charles Quairiaux[4], Eric B Taylor[7], Jared Rutter[8], José Manuel Nunes[9], Alan Carleton[4], Michael R Duchen[5], Carmen Sandi[2*], Jean-Claude Martinou[1*]

[1]Department of Cell Biology, University of Geneva, Geneva, Switzerland; [2]Laboratory of Behavioral Genetics, Ecole Polytechnique Fédérale de Lausanne, Lausanne, Switzerland; [3]Institut de Neurobiologie de la Méditerranée (INMED), Université d'Aix-Marseille, Marseille cedex, France; [4]Department of Basic Neuroscience, University of Geneva., Geneva, Switzerland; [5]Department of Cell and Developmental Biology, University College London, London, United Kingdom; [6]Cajal Institute, Madrid, Spain; [7]Department of Biochemistry and Fraternal Order of Eagles Diabetes Research Center, Carver College of Medicine, University of Iowa, Iowa City, United States; [8]Howard Hughes Medical Institute and Department of Biochemistry, University of Utah School of Medicine, Salt Lake City, United States; [9]Department of Genetic and Evolution, University of Geneva, Geneva, Switzerland

*For correspondence:
Marine.Laporte@unige.ch (MHL);
carmen.sandi@epfl.ch (CS);
Jean-Claude.Martinou@unige.ch (J-ClaudeM)

†These authors contributed equally to this work

Competing interest: The authors declare that no competing interests exist.

**Abstract** Neuronal excitation imposes a high demand of ATP in neurons. Most of the ATP derives primarily from pyruvate-mediated oxidative phosphorylation, a process that relies on import of pyruvate into mitochondria occuring exclusively via the mitochondrial pyruvate carrier (MPC). To investigate whether deficient oxidative phosphorylation impacts neuron excitability, we generated a mouse strain carrying a conditional deletion of MPC1, an essential subunit of the MPC, specifically in adult glutamatergic neurons. We found that, despite decreased levels of oxidative phosphorylation and decreased mitochondrial membrane potential in these excitatory neurons, mice were normal at rest. Surprisingly, in response to mild inhibition of GABA mediated synaptic activity, they rapidly developed severe seizures and died, whereas under similar conditions the behavior of control mice remained unchanged. We report that neurons with a deficient MPC were intrinsically hyperexcitable as a consequence of impaired calcium homeostasis, which reduced M-type potassium channel activity. Provision of ketone bodies restored energy status, calcium homeostasis and M-channel activity and attenuated seizures in animals fed a ketogenic diet. Our results provide an explanation for the seizures that frequently accompany a large number of neuropathologies, including cerebral ischemia and diverse mitochondriopathies, in which neurons experience an energy deficit.

## Editor's evaluation

This paper finds that conditional deletion in excitatory neurons of the only known pathway allowing the uptake of pyruvate into mitochondria, the mitochondrial pyruvate carrier, is tolerated in mice but produces hyperexcitability. The mice are more susceptible to epileptic seizures when inhibitory neurotransmission is impaired pharmacologically. Convincing evidence is presented that this

hyperexcitability is due to decreased activity of Kv7.2/7.3 channels, secondary to dysregulation of cellular calcium handling.

## Introduction

During neuronal activity, the energy demand imposed by neuronal excitation is mainly met by glucose, which is oxidized through glycolysis and oxidative phosphorylation (OXPHOS) to produce ATP (*Ashrafi and Ryan, 2017*). Oxidation of glucose starts in the cytosol and generates pyruvate, which, in turn, is transported into mitochondria through the mitochondrial pyruvate carrier (MPC) (*Bricker et al., 2012*; *Herzig et al., 2012*), where it fuels the TCA cycle and boosts OXPHOS. In neurons, pyruvate can also be generated through oxidation of astrocyte-derived lactate by the lactate dehydrogenase (*Pellerin et al., 2007*).

Besides energy production, glucose and pyruvate oxidation via the TCA cycle is also required for the synthesis of essential molecules, including the neurotransmitters glutamate and γ–aminobutyric acid (GABA). Therefore ATP production and neurotransmitter release are tightly linked to glucose and pyruvate metabolism. Accordingly, genetic pathologies linked to impaired glucose or pyruvate oxidation, such as mutations in the glucose transporter 1 (GLUT1) (*Brockmann et al., 2001*), pyruvate dehydrogenase (PDH) (*Ostergaard et al., 2009*), MPC (*Brivet et al., 2003*; *Oonthonpan et al., 2019*), or complexes of the respiratory chain (*Diaz et al., 2011*) result in severe synaptic dysfunction (*Beck and Yaari, 2008*). Not surprisingly, these diseases are associated with brain hypoactivity, although paradoxically they can be accompanied by neuronal hyperexcitability and behavioral seizures of varying severity. Indeed, this is paradoxical because it is generally thought that neuronal excitation imposes a high demand of OXPHOS-derived ATP in neurons. This raises the question of how these paroxysmal, ATP consuming events can occur in patients despite a global brain energy deficit.

A few years ago, the molecular identity of the MPC was revealed (*Bricker et al., 2012*; *Herzig et al., 2012*). The MPC is a heterodimer composed of two subunits, MPC1 and MPC2, inserted into the inner mitochondrial membrane (*Bricker et al., 2012*; *Herzig et al., 2012*). Deletion of MPC1 and MPC2 was sufficient to inactivate the carrier activity and its constitutive inactivation in the mouse caused embryonic lethality at E12 (*Vanderperre et al., 2016*; *Vigueira et al., 2014*). Interestingly, providing ketone bodies, which directly feed the TCA cycle with acetyl-CoA and boost OXPHOS, to the pregnant females allowed the embryos to survive until birth (*Vanderperre et al., 2016*).

Here, we hypothesized that downregulation of the MPC in neurons from adult mice would have a major impact on neuronal function and would result in decreased brain activity given the importance of the MPC in providing the TCA cycle with one of its main substrates. To test this hypothesis, we inactivated the MPC in adult mice, specifically in CamKIIα-expressing neurons (i.e. excitatory, glutamatergic neurons). We found that, under resting conditions, mice lacking MPC1 in these excitatory neurons were indistinguishable from control mice in their general exploratory, social and stress-coping behaviors. In response to inhibition of GABA mediated synaptic activity they developed far more severe seizures than controls. This phenotype was due to an intrinsic membrane hyperexcitability of MPC1-deficient glutamatergic neurons, which resulted from a calcium-mediated decrease in M-type K$^+$ channel activity. Strikingly, the hyperexcitability phenotype was reversed when the animals were maintained on a ketogenic diet.

## Results

### MPC-deficient cortical neurons display decreased pyruvate-mediated oxidative phosphorylation in vitro

To assess the role of the mitochondrial pyruvate carrier (MPC) in neuronal OXPHOS, we first used primary cultures of cortical neurons largely depleted of astrocytes (*Figure 1—figure supplement 1A*) and either RNA interference or pharmacological reagents to downregulate their MPC activity. To this end, two different shRNAs targeting MPC1 and three different pharmacological inhibitors of the carrier, Zaprinast (*Du et al., 2013*), Rosiglitazone (*Divakaruni et al., 2013*), and UK5099 (*Halestrap, 1975*) were used. Expression of either of the two shRNAs produced a significant reduction in MPC1 and MPC2 protein levels (the latter being unstable in the absence of MPC1) (*Figure 1—figure*

*supplement 1B, C*). Both genetic and pharmacological impairment of MPC activity resulted in decreased pyruvate-driven basal and maximal oxygen consumption rates (OCR) (*Figure 1A*, *Figure 1—figure supplement 1D*) and decreased mitochondrial ATP production (*Figure 1B*), which is consistent with previously published results (*Divakaruni et al., 2017*; *Grenell et al., 2019*). Furthermore, mitochondrial membrane potential, measured using mitotracker and TMRE was significantly reduced in MPC-deficient neurons (*Figure 1C* , F). This was associated with an increased extracellular acidification rate (*Figure 1—figure supplement 1E*) and increased glucose uptake, which was measured using the 2-NBDG import assay (*Figure 1—figure supplement 1F*), two hallmarks of aerobic glycolysis.

We have previously reported that ketone bodies can restore normal OXPHOS in MPC-deficient murine embryonic fibroblasts (*Vanderperre et al., 2016*). Consistent with this, we found here that addition of the ketone body β-hydroxybutyrate (βHB) (10 mM) to the culture medium rescued all observed defective functionalities in MPC-deficient neurons, including oxygen consumption, ATP production, membrane potential (*Figure 1D-F*) and both extracellular acidification rate and glucose uptake (*Figure 1—figure supplement 1G, H*). Thus, we conclude that MPC-deficient neurons display low pyruvate-mediated oxidative phosphorylation and high aerobic glycolysis, both reversed with βHB.

## Generation of mice with inducible MPC1 gene deletion in adult glutamatergic neurons

Based on the results described above, and because neural excitation requires massive levels of ATP, we hypothesized that loss of MPC activity would reduce excitability especially in glutamatergic neurons that are high energy consumers. To test this hypothesis, we generated a mouse strain with an inducible deletion of the MPC1 gene, specifically in the $Ca^{2+}$-calmodulin kinase IIα (CamKIIα)-expressing neurons, found predominantly in the hippocampus and cortex (*Wang et al., 2013*). We crossed *Mpc1*$^{flox/flox}$ mice with the commercially available *CamkIIa*$^{CreERT2}$ mice (*Figure 2A*). Induction of Cre activity by injection of Tamoxifen for five consecutive days resulted in deletion of MPC1 FLOXed alleles in the CamKIIα-expressing adult neurons (*Figure 2A*). Hereafter, we refer to these mice as neuro-MPC1-KO. In situ immunofluorescence analyses showed a decrease in neuronal MPC1 immunostaining in various layers of the cortex of neuro-MPC1-KO mice (*Figure 2B*). Western blot analysis of whole cortex, synaptosomes and mitochondria showed a significant decrease of both MPC1 and MPC2 in neuro-MPC1-KO mice compared to neuro-MPC1-WT mice (*Figure 2C*). Consistent with the results obtained in cultured neurons, we found that synaptosomes prepared from the cortex of neuro-MPC1-KO mice displayed lower oxygen consumption and imported higher amounts of glucose compared to synaptosomes from neuro-MPC1-WT mice (*Figure 2—figure supplement 1A, B*). Importantly, the lack of MPC1 did not affect the neuronal cell survival quantified either by counting the total number of cells, or by the number of apoptotic (TUNEL positive) cells (*Figure 2—figure supplement 1C, D*). At adulthood, both genotypes displayed similar body weight and fat mass composition (*Figure 2—figure supplement 1E, F*). At the behavioral level, adult neuro-MPC1-KO mice showed a tendency toward lower anxiety-like behaviors, but no difference in general locomotion, sociability or stress-coping behaviors (*Figure 2—figure supplement 1G, H*).

These data indicate that, under resting conditions, the excitatory neurons in most adult mice have the ability to bypass the MPC to meet their metabolic demands.

## Neuro-MPC1-KO mice are highly sensitive to pro-convulsant drugs and develop acute epileptic-like seizures

The output activity of a neuron results from the balance between the excitatory and the inhibitory inputs it receives. Perturbation of this delicate balance can lead to severe seizures as a result of exacerbated, uncontrolled neuronal firing. To test whether OXPHOS-deficient excitatory neurons could sustain intense neuronal firing, we challenged neuro-MPC1-KO adult mice, with either pentylenetetrazole (PTZ), a GABA receptor antagonist, or kainic acid, an activator of glutamate receptors. We used the PTZ kindling protocol described previously (*Dhir, 2012*), in which a sub-convulsant dose (35 mg/kg) of PTZ is injected intraperitoneally (ip) once every 2 days on a period of 15 days (*Figure 3A*). Phenotypic scoring after each PTZ injection in neuro-MPC1-WT mice showed a progressive sensitization (kindling) starting with hypoactivity after the first injection (scored as 1); a few brief and transient muscle contractions (jerks, scored as 2) or appearance of tail rigidity (Straub's tail, scored as 3) following

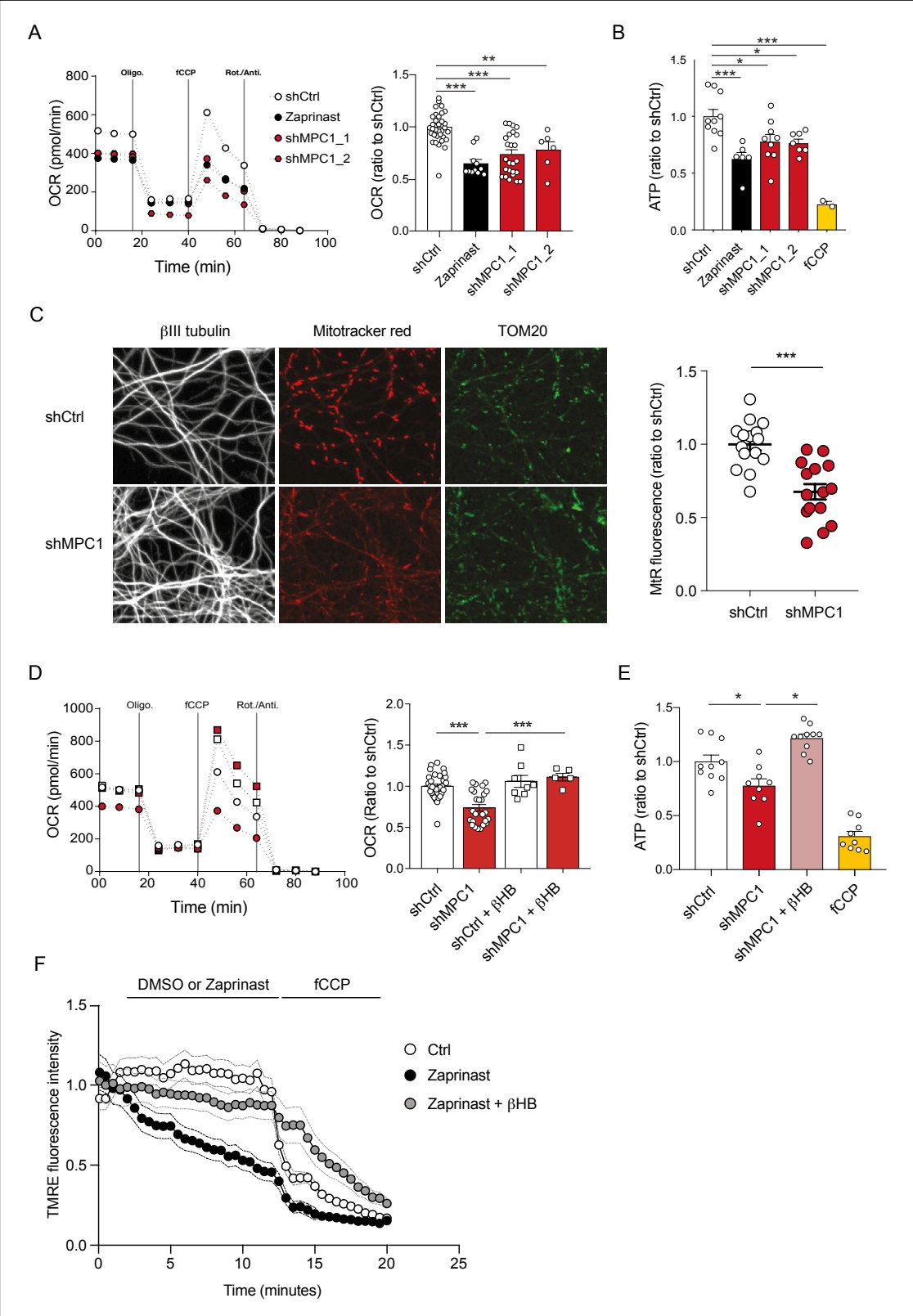

**Figure 1.** MPC-deficient neurons display defects in mitochondrial respiration and membrane potential. (**A**) Profile and quantification of oxygen consumption rates (OCR) cortical neurons expressing either shCtrl, or shMPC1_1 and shMPC1_2 for 7 days, or in the presence of Zaprinast (5 µM, 1 hr). Data were obtained using the Seahorse XF analyzer. Assays were performed in the presence of pyruvate (5 mM) and glucose (5 mM) as carbon sources. Quantification of basal OCR is expressed as ratio of ShCtrl. N = 10,7,9,7 and 2 independent experiments. N = 33,11,25 and 6 independent

*Figure 1 continued on next page*

*Figure 1 continued*

experiments. One-way ANOVA + Tukey's post-hoc test (shCtrl vs Zaprinast p = 0.0001, shCtrl vs shMPC1_1 p = 0.0001, shCtrl vs shMPC1_2 p = 0.0013). (**B**) ATP content in MPC-deficient cortical neurons treated with either shCtrl, Zaprinast or shMPC1_1 and shMPC1_2. fCCP (4 µM) treatment reveals the non-mitochondrial ATP. N = 10,7,9,7 and 2 independent experiments. One-way ANOVA + Tukey's post-hoc test (shCtrl vs Zaprinast p = 0.0006, shCtrl vs shMPC1_1 p = 0.0223, shCtrl vs shMPC1_2 p = 0.0242, shCtrl vs fCCP p = 0.0001). (**C**) Mitochondrial membrane potential of MPC-deficient cortical neurons. Neurons were incubated with Mitotracker red (MtR) (1 µM) prior fixation, immunostained for βIII tubulin (neuron) and TOM20 (mitochondria). Quantification of Mitotracker red fluorescence in each βIII tubulin-positive cell (red) was reported to TOM20 signal (green). N = 15 neurons from three independent experiments. Unpaired t test (shCtrl vs shMPC1 p = 0.0001). (**D**) Profile and quantification of oxygen consumption rates (OCR) in cortical neurons expressing shCtrl or shMPC1_1 for 7 days. Data were obtained using the Seahorse XF analyzer. Assays were performed in the presence of pyruvate (5 mM) and glucose (5 mM) as carbon sources + 10 mM βHB when indicated. Quantification of basal OCR is expressed as ratio of control condition shCtrl. N = 12 independent experiment. One-way ANOVA + Holm Sidak's post-hoc test (shCtrl vs shMPC1 p = 0.0001, shMPC1 vs shMPC1+βHB p = 0.0002). (**E**) ATP content in MPC-deficient cortical neurons treated with shCtrl or shMPC1 in presence or absence of 10 mM βHB. fCCP (4 µM) treatment reveals the non-mitochondrial ATP. N = 10, 9, 10, 9 independent experiments. One-way ANOVA + Holm Sidak's post-hoc test (shCtrl vs shMPC1 p = 0.0145, shMPC1 vs shMPC1+βHB p = 0.0143, shCtrl vs fCCP p = 0.0001). (**F**) Cortical neurons were incubated with TMRE (50 nM) +/- βHB (10 mM) for 30 min and recorded by live microscopy. Neurons were incubated with DMSO or Zaprinast (5 µM) 2.5 min after the beginning of the acquisition and recorded for 5 min prior fCCP injection. N = 15 independent experiments. One-way ANOVA + Holm Sidak's post-hoc test (shCtrl vs Zaprinast p = 0.0028, Zaprinast vs Zaprinast+βHB *P* = 0.0008).

The online version of this article includes the following figure supplement(s) for figure 1:

**Figure supplement 1.** Effects of MPC inhibition in cortical neurons in vitro.

the second or third injection; and convulsive status epilepticus (scored as 6) after the 6th or 7th injection (*Figure 3A*, B). In contrast, all neuro-MPC1-KO mice developed severe, prolonged seizures (score 6) within 10 min of the first PTZ injection (*Figure 3B*) and all died during seizures within the next three PTZ injections (*Figure 3—figure supplement 1A*). When mice were injected with 20 mg/kg kainic acid intraperitoneally, a similar hypersensitivity (score 6) was observed in neuro-MPC1-KO mice indicating that this sensitivity was not restricted to PTZ (*Figure 3—figure supplement 1B*).

In a parallel series of experiments, and in order to assess the specificity of our results to excitatory neurons, we investigated the effects of PTZ in mice in which MPC1 was deleted in adult astrocytes (hereafter termed astro-MPC1-KO mice) (*Figure 3—figure supplement 1C-E*). In contrast to neuro-MPC1-KO mice, astro-MPC1-KO mice showed the same response as control animals following PTZ injection (*Figure 3—figure supplement 1F-G*), indicating that the phenotype observed in neuro-MPC1-KO mice is linked to the deletion of MPC1 in excitatory neurons.

To characterize the seizure symptoms in more detail, we recorded the electrical activity in the brains of neuro-MPC1-WT and neuro-MPC1-KO mice by electroencephalogram (EEG) following a single injection of PTZ (*Figure 3C*). In neuro-MPC1-KO mice, rhythmic EEG patterns emerged within 5–10 min after PTZ injection, invading all electrodes (*Figure 3C*). These electrical patterns coincided with the occurrence of behavioral manifestations of seizures, that is tonic-clonic movements. Rapidly thereafter, large spike and wave discharges developed, again invading all surface electrodes and coinciding with numerous fast ripples (*Figure 3C*, **inset**). Such EEG patterns are characteristic of seizure episodes in humans and were not observed in the PTZ-injected neuro-MPC1-WT mice. These data indicate that neuro-MPC1-KO mice develop an epilepsy-like phenotype following administration of a single sub-convulsant dose of PTZ.

We also tested whether we could reproduce the seizure phenotype using hippocampal organotypic cultures from *CamkIIa^CreERT2^-Mpc1^flox/flox^* mice exposed to PTZ, combined with calcium imaging. Individual neurons in hippocampal slices from both WT and KO mice exhibited spontaneous calcium activity throughout the duration of the recordings (*Figure 3D*, E and *Video 1*, *Video 2*) although, interestingly, the frequency of calcium events, as well as the number of co-activation events (i.e. neuronal synchronizations above chance levels) generated in MPC1-deficient neurons were significantly higher than those generated in MPC1-WT neurons (*Figure 3F*, G). In contrast, neither the amplitude nor the duration of the discharges was modified (*Figure 3H*, I). These results suggest that neuro-MPC1-KO neurons are more active and are more often recruited into synchronized patterns associated with the epileptic activity.

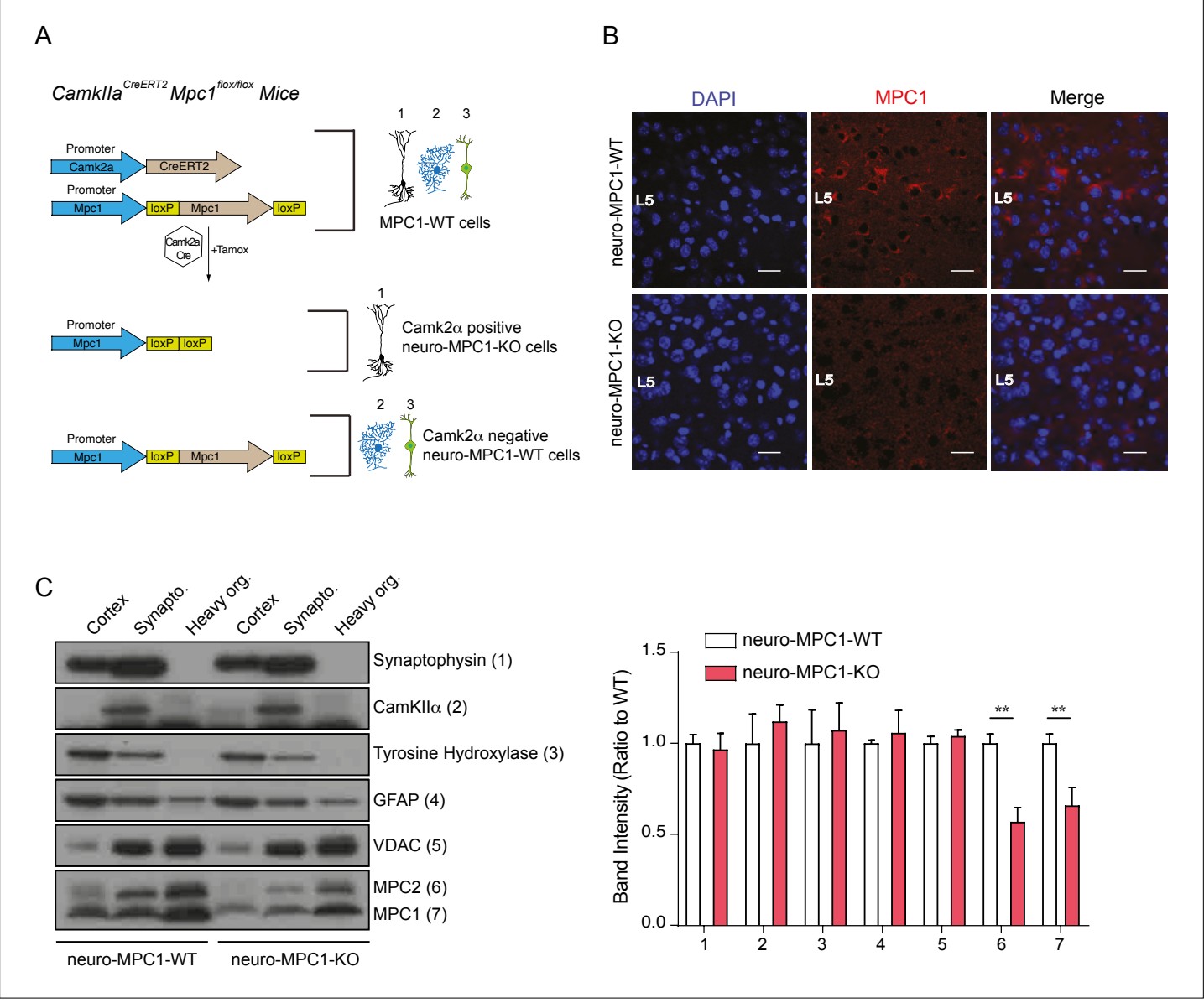

**Figure 2.** Generation of mice with an inducible deletion of the MPC1 gene in adult glutamatergic neurons. (**A**) Strategies used to generate *CamKIIa^{CreERT2}-Mpc1^{flox/flox}* mice. Upon Tamoxifen injection, expression of the Cre recombinase in CamKIIα glutamatergic neurons drives deletion of the MPC1 gene. These mice are referred to as neuro-MPC1-KO or neuro-MPC1-WT when they are Cre- (1. Glutamatergic neuron; 2. Astrocytes; 3. Inhibitory neuron). (**B**) Immunostaining of MPC1 (red) in cortical sections from neuro-MPC1-WT and neuro-MPC1-KO mice (scale bar: 100 µm). (**C**) Western blot analysis of whole cortex, synaptosome lysates and heavy organelles (mainly mitochondria), obtained from brains of neuro-MPC1-WT and neuro-MPC1-KO mice using neuronal (Synaptophysin, tyrosine hydroxylase, CamKIIα) and astroglial markers (GFAP) as well as mitochondrial markers (MPC1, MPC2 and VDAC). Note that synaptosomes are enriched for CamKIIα, a marker of excitatory neurons. Quantification (right panel) shows that except for MPC1 and MPC2, the content of these markers is similar in WT and KO preparations. N = 6 independent neuro-MPC1-WT and neuro-MPC1-KO mice. Mann-Whitney test ((6) neuro-MPC1-WT vs neuro-MPC1-KO p = 0.0286, (7) neuro-MPC1-WT vs neuro-MPC1-KO p = 0.0152).

The online version of this article includes the following figure supplement(s) for figure 2:

**Figure supplement 1.** Metabolic effects of MPC deletion in excitatory neurons in vitro and behavioural consequences in neuro-MPC1-WT and KO animals.

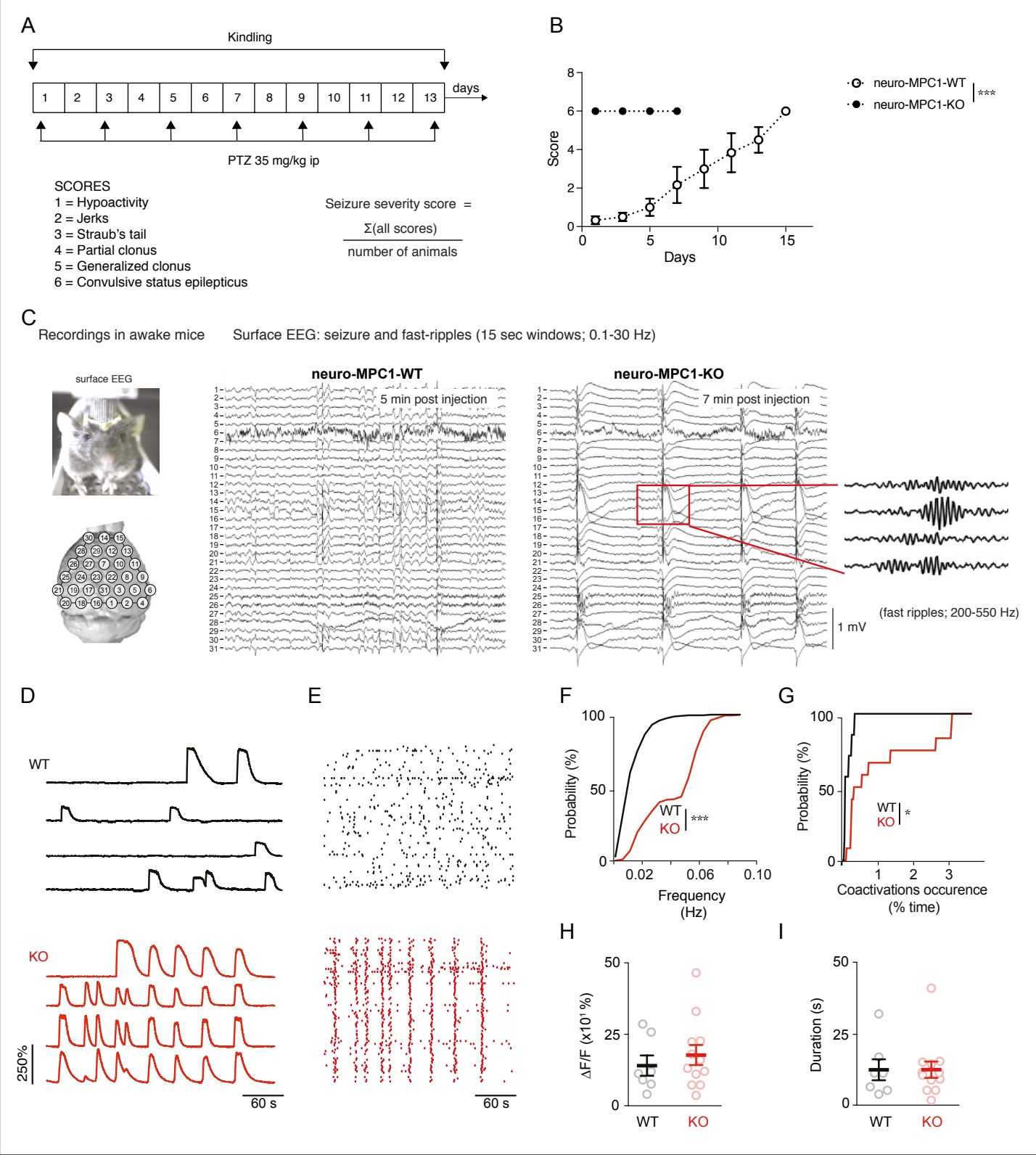

**Figure 3.** Neuro-MPC1-KO mice are highly sensitive to pro-convulsant drugs and develop acute epileptic-like seizures. (**A**) Schematic description of the PTZ kindling protocol. (**B**) Seizure severity scores reflecting the different clinical symptoms as indicated, obtained for neuro-MPC1-WT or neuro-MPC1-KO. N = 8 independent neuro-MPC1-WT and neuro-MPC1-KO mice. Two way ANOVA (F(7,70)=19, p = 0.0001). (**C**) Illustration of the recording setups in awake mice indicating the position of surface EEG electrodes and representative example of a seizure recorded in a neuro-MPC1-KO mouse after

*Figure 3 continued on next page*

*Figure 3 continued*

injection of 35 mg/kg PTZ during surface EEG recordings. The inset shows an example of fast ripples generated during an ictal epileptic discharge. (**D–I**) GCaMP6S calcium imaging of the CA1 area from hippocampal slices in the presence of Carbachol (50 μM) and PTZ (2 mM). Slices were prepared from WT animals (top, black) or from KO animals with no pre-treatment (bottom, red). (**D**) $Ca^{2+}$ sweeps recorded in four representative GCaMP6S-expressing neurons. (**E**) Raster plots of $Ca^{2+}$ transient onsets extracted from all recorded neurons in a given slice. (**F**) Cumulative distribution of the frequency of the calcium events in all the recorded neurons. N = 7, 12 independent experiments. Kolmogorov-Smirnov test (WT vs KO p = 0.0001). (**G**) Cumulative distribution of the occurrence of neuronal co-activations exceeding chance levels as a function of time N = 7, 12 independent experiments. Kolmogorov-Smirnov test (WT vs KO p = 0.0344). Amplitude (**H**), and duration (**I**) of the calcium events recorded in all neurons of the hippocampus. N = 7, 12 independent experiments. Mann-Whitney test (Amplitude: WT vs KO p = 0.5918; Duration: WT vs KO p = 0.9182).

The online version of this article includes the following figure supplement(s) for figure 3:

**Figure supplement 1.** Generation and phenotype of Astro-MPC1-KO mice.

## Inhibition of PTZ-induced seizures in neuro-MPC1-KO mice by the ketogenic diet

The ketogenic diet (KD) has been reported to decrease seizures in patients with pharmacologically refractory epilepsy (*Carroll et al., 2019*). Ketone bodies, mainly generated by the liver during fasting and hypoglycaemia, are used by neurons to provide the TCA cycle with acetyl-CoA, normally provided by pyruvate dehydrogenase-mediated oxidation of pyruvate. Thus, ketone bodies ensure that oxidative phosphorylation and ATP production is maintained in neurons in conditions of glucose starvation. We tested whether a ketogenic diet could prevent PTZ-induced seizures in neuro-MPC1-KO mice. As previously reported (*Wirrell et al., 2018*), we found that the KD produces a decrease in glycaemia and an increase in the blood level of 3-β-hydroxy-butyrate (βHB), one of the three major ketone bodies generated by the liver (*Figure 4—figure supplement 1A, B*). In addition, we found that mice fed on the KD for 1 week were completely resistant to PTZ injection (*Figure 4A*). Supplementing the drinking water with 1% βHB was sufficient to prevent PTZ-induced seizures (*Figure 4B*). Similarly, ip administration of βHB (1 g/Kg) 15 min before PTZ injection, or starvation overnight, both of which conditions led to increased βHB blood levels (*Figure 4—figure supplement 1C-E*), significantly reduced the PTZ-induced clinical score of neuro-MPC1-KO mice (*Figure 4B*, C). Similarly, supplementing the drinking water with 1% acetoacetate, another type of ketone body, reduced PTZ-induced seizures in neuro-MPC1-mice (*Figure 4D*). These results indicate that the phenotype displayed by the neuro-MPC1-KO mice is mainly metabolic in origin and is unlikely to be the consequence of neuronal network remodeling.

## MPC1-deficient neurons display intrinsic hyperexcitability, which is prevented by ketone bodies

To investigate the cellular mechanisms that might mediate the sensitivity of neuro-MPC1-KO mice to pro-convulsant drugs, we examined the electrophysiological properties of MPC1-deficient neurons. To this end, we performed whole-cell patch clamp recordings in acute hippocampal slices from neuro-MPC1-KO mice and their neuro-MPC1-WT littermates. CA1 pyramidal cells from neuro-MPC1-KO mice exhibited higher discharge frequency compared to neurons from neuro-MPC1-WT mice when firing was elicited by somatic injections of current ramps of increasing amplitude (*Figure 5A*, B). Neurons

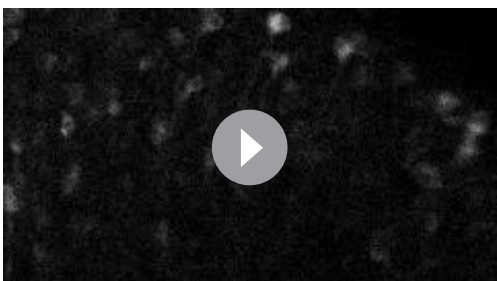

**Video 1.** Spontaneous calcium activity in neuro-MPC1-WT hippocampal slices following PTZ addition.
https://elifesciences.org/articles/72595/figures#video1

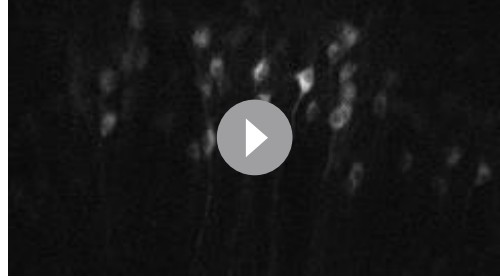

**Video 2.** Spontaneous calcium activity in neuro-MPC1-KO hippocampal slices following PTZ addition.
https://elifesciences.org/articles/72595/figures#video2

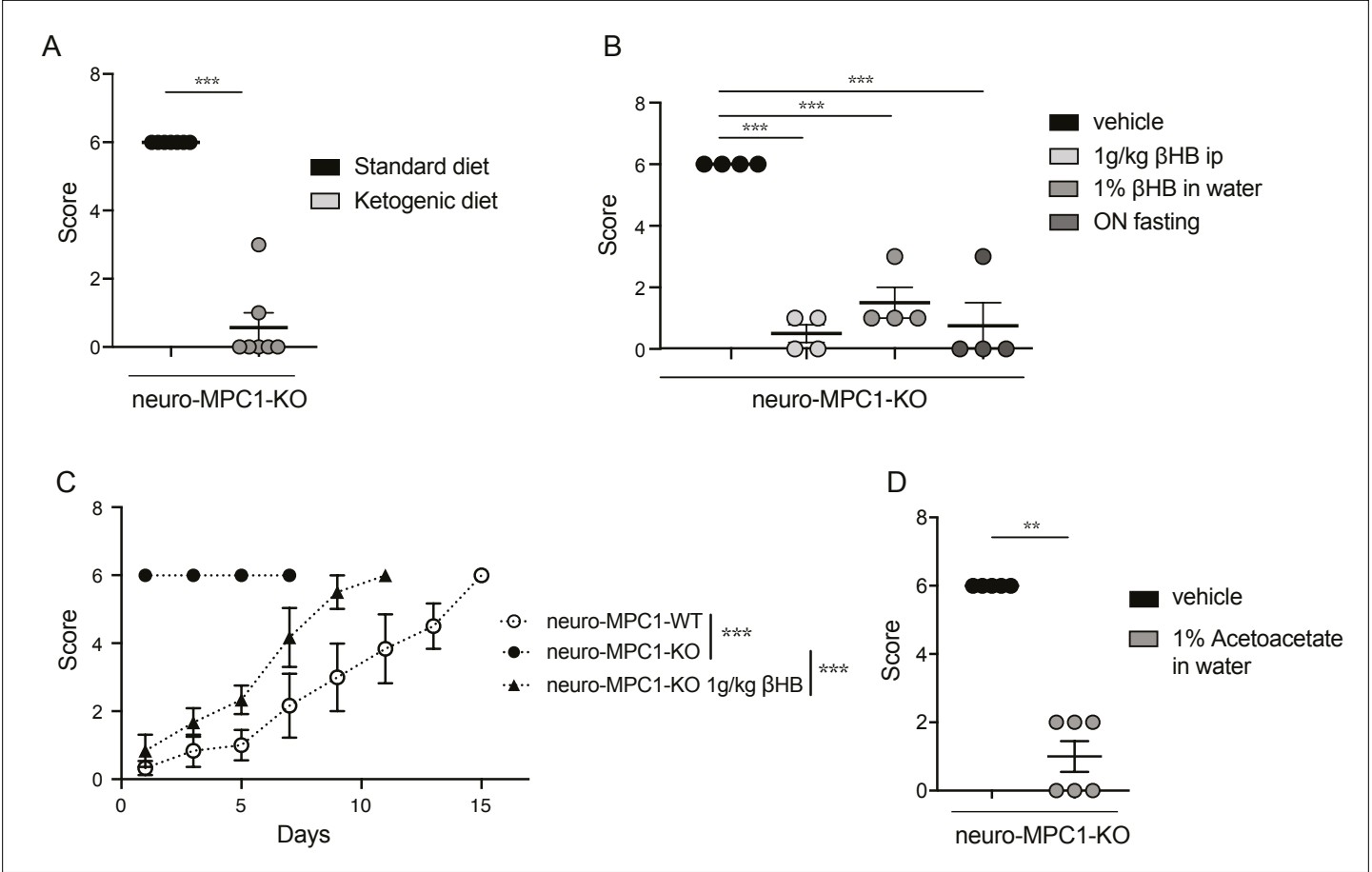

**Figure 4.** Ketogenic diet prevents the epileptic phenotype of neuro-MPC1-KO mice. (**A**) Effect of the ketogenic diet (KD) on PTZ-induced seizure. All neuro-MPC1-KO mice were maintained on the Standard (SD) or ketogenic (KD) diet for 7 days prior to challenge with a single dose of PTZ. Clinical scores were assessed directly following injection. N = 7 independent neuro-MPC1-KO mice. Mann-Whitney (neuro-MPC1-KO SD vs neuro-MPC1-KO KD p = 0.0008). (**B**) Effects 1% βHB in the drinking water for 7 days, overnight fasting or ip injection of βHB 15 min before administration of PTZ into neuro-MPC1-KO mice. N = 4 independent neuro-MPC1-KO mice. One-way ANOVA + Holm Sidak's post-hoc test (Vehicle vs all conditions p = 0.0001). (**C**) Effect of βHB on PTZ-induced seizure: mice were injected ip with 1 g/kg βHB, 15 min before each PTZ injection and scored for clinical symptoms. N = 6 independent mice. Two-way ANOVA + Holm Sidak's post-hoc test (F(10, 75) = 8, Neuro-MPC1-WT vs neuro-MPC1-KO, neuro-MPC1-KO vs neuro-MPC1-KO + βHB p = 0.0001). (**D**) Effect of acetoacetate (1% in drinking water) for 7 days on PTZ-induced seizures. Clinical scores were assessed directly following injection. N = 5 and 6 mice for vehicle or acetoacetate IP injected mice, respectively. Mann-Whitney (neuro-MPC1-KO vehicle vs neuro-MPC1-KO Acetoacetate p = 0.0022).

The online version of this article includes the following figure supplement(s) for figure 4:

**Figure supplement 1.** Glycemia and ketonemia measurements in neuro-MPC1-WT and neuro-MPC1-KO mice fed on a ketogenic diet.

from neuro-MPC1-KO mice required less current injection (rheobase, *Figure 5C*) to reach the firing threshold, which was more hyperpolarized when compared to neuro-MPC1-WT cells (*Figure 5D*). Similarly, MPC1-KO neurons displayed higher firing when depolarization was induced with squared current pulses (*Figure 5—figure supplement 1A, B*).

Next, we asked whether ketone bodies, which as shown in *Figure 4* prevent PTZ-induced seizures, could modulate neuronal excitability and restore normal cell discharges in neuro-MPC1-KO mice. For these experiments, we first recorded action potential firing under control conditions, and then perfused the slices with βHB (2 mM, > 20 min exposure) (*Ma et al., 2007*). As shown in *Figure 5*, whereas cell firing was unaltered in neuro-MPC1-WT cells (*Figure 5E*, F), βHB reduced excitability in pyramidal cells from the neuro-MPC1-KO mice (*Figure 5G*, H). Control experiments showed that cell excitability from both genotypes was unchanged during prolonged recordings (*Figure 5F*, H), confirming that the change in neuro-MPC1-KO firing was not due to a rundown in cellular excitability caused by, for example, cell dialysis.

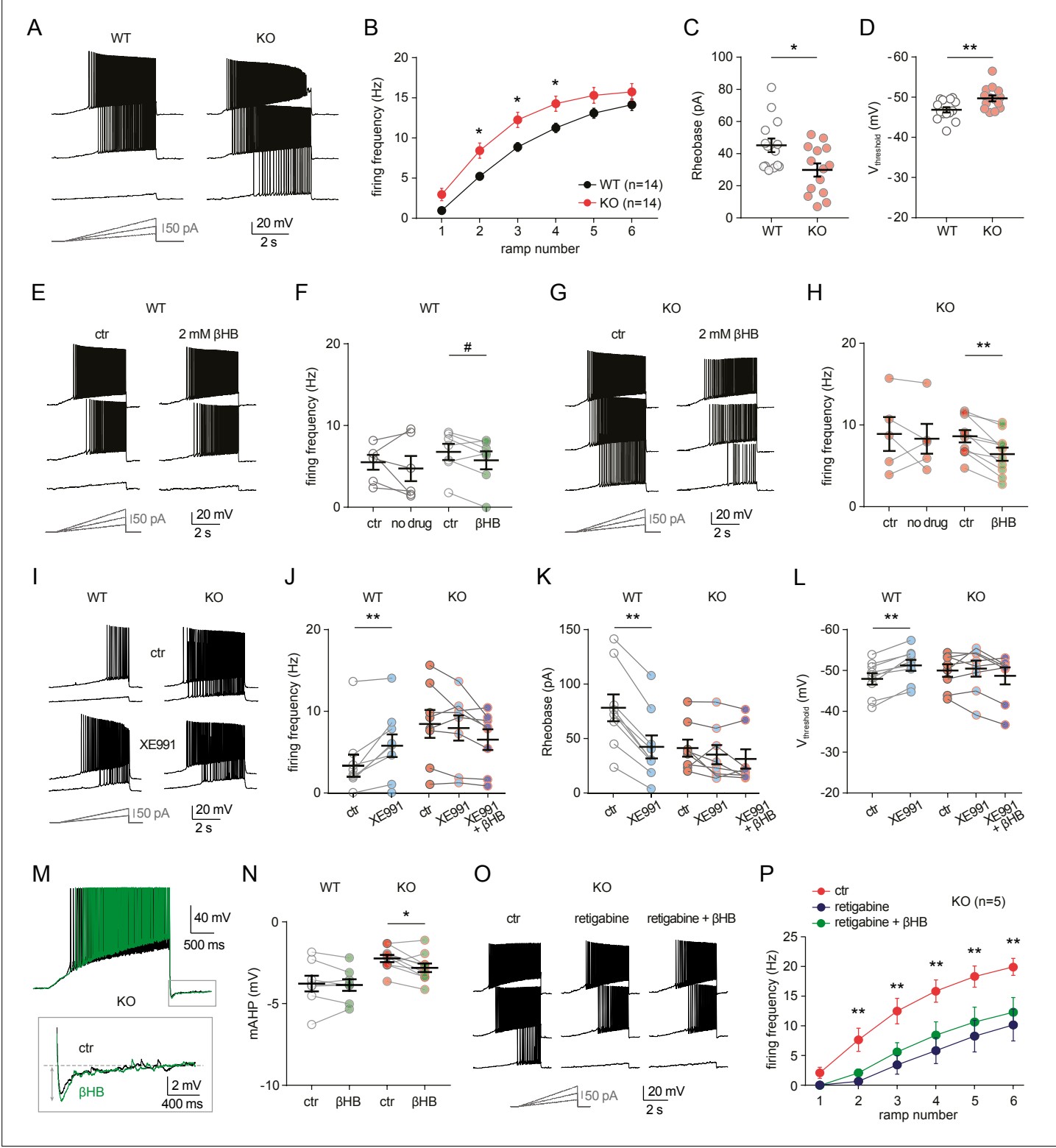

**Figure 5.** MPC1 deletion increases intrinsic excitability in CA1 pyramidal cells. (**A**) Example voltage responses elicited in CA1 pyramidal cells from wild-type (WT) and neuro-MPC-KO (KO) by injection of current ramps (protocol at the bottom, only three of six ramps displayed). (**B**) Frequency-current (F–I) relationship of action potential discharges, indicating higher spiking frequency in KO cells (Two-way ANOVA, $F_{(1, 156)}$ 33.43, p < 0.0001). (**C**) The rheobase was reduced in KO cells (Mann-Whitney test, U = 53.5, p = 0.0406). (**D**) KO cells exhibited more hyperpolarized threshold potential (unpaired t test, t = 2.856, p = 0.0084). (**E**) Example traces showing lack of significant changes in WT cell firing after bath application of the ketone body

*Figure 5 continued on next page*

*Figure 5 continued*

β-hydroxybutyrate (βHB, 2 mM, > 20 min exposure). (**F**) Average firing frequency elicited by the 3rd ramp of current injections in WT cells in control condition (5 min after whole-cell establishment) and after 20 min of either no drug exposure or βHB application (no drug: paired t test, t = 0.664, p = 0.5362; βHB: paired t test, t = 2.1, p = 0.0804). (**G**), (**H**) Example traces and summary graphs indicating significant reduction in KO cell firing after βHB application (no drug: paired t test, t = 0.4691, p = 0.6634; βHB: paired t test, t = 5.339, p = 0.0005). (**I**) Example traces of cell firing in control and after bath application of XE991 (10 μM) in WT and KO neurons. (**J**) Average firing frequency elicited by the 2nd ramp of current injections in control and after XE991 application, indicating increased excitability in WT cells (paired t test, t = 3.735, p = 0.0057). XE991 was ineffective in KO cells, in which subsequent application of βHB also failed to modulate excitability (One-way ANOVA, $F_{(1.69, 11.89)}$ = 4.76, p = 0.0347, Holm-Sidak's multiple comparison p > 0.05). (**K**) XE991 significantly reduced the rheobase of WT cells (paired t test, t = 11, p < 0.001), but not of KO cells, in which subsequent βHB application was also ineffective (One-way ANOVA, $F_{(1.785, 12.5)}$ = 2.99, p = 0.091). (**L**) XE991 induced a shift in the threshold potential of WT cells (paired t test, t = 6.001, p = 0.0003), but did not affect KO cells, in which subsequent βHB application was also ineffective (One-way ANOVA, $F_{(1.812, 12.68)}$ = 1.78, p = 0.209). (**M**) Example traces of KO cell firing elicited by a current ramp (300 pA max amplitude, APs are trimmed) in control and after βHB exposure, with expanded portion at the bottom indicating mAHP measurement. **N**) Summary graph of mAHP values in WT and KO cells in control and after βHB exposure, indicating significant increase in KO (unpaired t test, t = 2.89, p = 0.0179. (**O**) Example traces of KO cell firing before and after application of retigabine (10 μM), and subsequent βHB superfusion (2 mM). (**P**) F-I relationships in KO cells, indicating reduced spiking frequency after retigabine application, with no additional effect of βHB (Two-way repeated measures ANOVA, $F_{(2, 48)}$ = 89.15, p < 0.0001).

The online version of this article includes the following figure supplement(s) for figure 5:

**Figure supplement 1.** CA1 pyramidal cells from neuro-MPC1-KO mice exhibit altered intrinsic excitability and membrane properties but no changes in glutamatergic transmission.

Taken together, these results indicate that ketone bodies reduce the intrinsic hyperexcitability of glutamatergic cells from neuro-MPC1-KO mice, providing a plausible explanation for the protective effect of the KD against PTZ-induced seizures.

## MPC1-deficient neurons display altered M-type potassium channel activation, which is corrected by ß-hydroxybutyrate

To gain insight into the mechanisms governing neuronal hyperexcitability, we analysed the cellular passive properties and action potential characteristics of all recordings performed in cells from neuro-MPC1-KO and neuro-MPC1-WT mice (*Figure 5—figure supplement 1C-K*). The reduction in rheobase and the shift in threshold potential induced by MPC1 deletion were accompanied by several changes in passive and active membrane properties governing cell excitability, including a significant increase in the input resistance ($R_i$) and in the voltage response to a depolarizing current injection (depol$_{sub}$), along with a marginally significant reduction in HCN channel-mediated sag (*Figure 5—figure supplement 1C-I*). The fast afterhyperpolarization (fAHP) accompanying action potentials was not altered, ruling out a major contribution of BK channels (*Figure 5—figure supplement 1J*). However, the medium afterhyperpolarization (mAHP), measured as the negative peak of the voltage deflection at the offset of the depolarizing ramps was significantly reduced in cells from neuro-MPC1-KO mice (*Figure 5—figure supplement 1K*). In CA1 pyramidal cells, mAHP is primarily mediated by the activation KCNQ2/3 (Kv7.2 and Kv7.3) channels, which generate an M-type K$^+$ conductance regulating intrinsic excitability and synaptic integration (*Gu et al., 2008*; *Peters et al., 2005*). Opening of these channels produces an outward potassium current that functions as a 'brake' for neurons receiving persistent excitatory input (*Greene and Hoshi, 2017*). Consistently, mutations in KCNQ2/3 genes have been associated with seizures in the mouse (*Singh et al., 2008*), as well as in patients (*Jentsch et al., 2000*), pointing to these channels as interesting targets for anticonvulsant therapy (*Barrese et al., 2018*). To test whether neuro-MPC1-KO mice displayed an altered contribution of the M-type K$^+$ conductance, we tested the effect of the M-type channel blocker XE991 (10 μM) on CA1 pyramidal cell firing. XE991 led to a significant increase in firing frequency of neuro-MPC1-WT cells, whereas firing of neuro-MPC1-KO cells was not significantly modified (*Figure 5I and J*). Consistently, XE991 induced a significant reduction in the rheobase and a shift in the threshold potential in neuro-MPC1-WT cells, but had no impact on neuro-MPC1-KO cells (*Figure 5K and L*), pointing to a limited activity of KCNQ2/3 channels in these neurons. Interestingly, bath application of βHB following KCNQ2/3 channel blockade with XE991 failed to reduce the hyperexcitability of neuro-MPC1-KO mice (*Figure 5J–L*). We also noticed that, in the absence of XE991, the reduction of intrinsic excitability by βHB in MPC-deficient neurons was accompanied by a significant increase in mAHP (*Figure 5M*, N), suggesting that βHB may potentiate the recruitment of the M-type K$^+$ channels. Moreover, the M-type

channel activator retigabine (10 µM) effectively decreased the hyperexcitability of pyramidal cells from neuro-MPC1-KO mice to a level that was no further affected by βHB (*Figure 5O*, P). This suggests that βHB and retigabine display a similar mechanism of action, which is consistent with recent findings showing that βHB can directly bind to and activate KCNQ2/3 channels (*Manville et al., 2020*).

We finally tested whether the increased neuronal excitability in neuro-MPC1-KO mice was also accompanied by alterations in glutamatergic transmission. In acute slices, we recorded field potentials in CA1 stratum radiatum elicited by electrical stimulation of the Schaffer collaterals (*Figure 5—figure supplement 1L*). No overt genotype differences were found in the input-output curves of field excitatory postsynaptic potentials (fEPSPs), and the lack of changes in paired-pulse ratio indicated no major alteration in the presynaptic release (*Figure 5—figure supplement 1M, N*).

Altogether, these results indicate that the hyperexcitability of CA1 pyramidal neurons from neuro-MPC1-KO mice is mediated by alterations in intrinsic cell excitability associated with a reduced M-type K$^+$ channel activation, with no major changes in excitatory synaptic inputs.

## Alteration of calcium homeostasis in MPC1-deficient neurons

The conductance of KCNQ channels is regulated by phosphatidylinositol-4,5-bisphosphate (PIP2) and calmodulin (CaM) (*Alaimo and Villarroel, 2018*; *Gamper and Shapiro, 2003*). In particular, reduction in free CaM in hippocampal neurons decreases M-current density and increases neuronal excitability (*Shahidullah et al., 2005*; *Zhou et al., 2016*). Thus, calcium could trigger loss of interaction of CaM and KCNQ2/3 channels, leading to M-type current suppression (*Kosenko and Hoshi, 2013*).

We tested whether disruption of calcium homeostasis could be responsible for the deficit in the M-type K$^+$ channel activity displayed by MPC1-deficient neurons. We first assessed whether calcium homeostasis was perturbed in MPC-deficient cortical neurons in vitro. Using the fluorimetric calcium probes Fura2-AM and the low affinity FuraFF-AM combined with live cell imaging, we found a significant increase in the peak concentration of cytosolic calcium upon depolarization of both control and MPC-deficient neurons in response to either 10 µM glutamate (*Figure 6A*, B; *Figure 6—figure supplement 1*) or 50 mM KCl (*Figure 6C–E*). However, while the peak of calcium concentration was transient in control neurons, and returned to basal levels, both the magnitude and duration of the calcium elevation were greater in MPC-deficient neurons (*Figure 6A and C*). Interestingly, the long lasting increased calcium level in MPC-deficient neurons was abolished by addition of 10 mM βHB to the culture medium 30 min prior to recording (*Figure 6A*, B, *Figure 6—figure supplement 1C*). Together these results show that loss of MPC activity leads to a significant increase of cytosolic calcium levels in depolarized neurons.

Mitochondria import calcium through the mitochondrial calcium uniporter (MCU) in a membrane potential dependent manner and thereby play a major role in calcium homeostasis (*Giorgi et al., 2012*). Given that the mitochondrial membrane potential of the MPC-deficient neurons was reduced (*Figure 1C*, F), we assessed whether the increased cytosolic calcium levels in MPC-deficient neurons could result from impaired calcium uptake by mitochondria, similar to what would be expected for reduced MCU activity. We monitored cytosolic calcium in cultured cortical neurons at rest or upon stimulation with 50 mM KCl, in the presence or absence of chemical inhibitors of the MPC. The MCU activity was downregulated either using the pharmacological inhibitor RU360 (*Márta et al., 2021*) or RNA interference (*Figure 6—figure supplement 2A*). These experiments showed that upon neuron depolarization with 50 mM KCl, the cytosolic calcium level was significantly higher in neurons in which MPC or MCU had been inactivated (*Figure 6C–E*; *Figure 6—figure supplement 2B*). The level of cytosolic calcium was not further increased when both MCU and MCP were inactivated concomitantly (*Figure 6—figure supplement 2B-D*). These results confirmed that impairing the MCU, similar to inhibiting the MPC, resulted in increased cytosolic calcium levels in depolarized neurons.

To pursue our investigations further, we used two strategies to assess mitochondrial calcium: (i) an indirect approach consisting in measuring the difference in the signal emitted by the cytosolic calcium probe, before and after depolarization of mitochondria with fCCP. Upon depolarization, mitochondria are expected to release their calcium content. (ii) a direct approach using the calcium luminescent probe aequorin targeted to mitochondria (*Bonora et al., 2013*; *Tosatto et al., 2016*). As expected, fCCP-induced mitochondrial uncoupling resulted in an increased signal from the cytosolic calcium probe in control neurons and in neurons with inactive MPC or MCU. However, the difference in signal intensity, before and after fCCP addition, was significantly attenuated in MPC and/or MCU-deficient

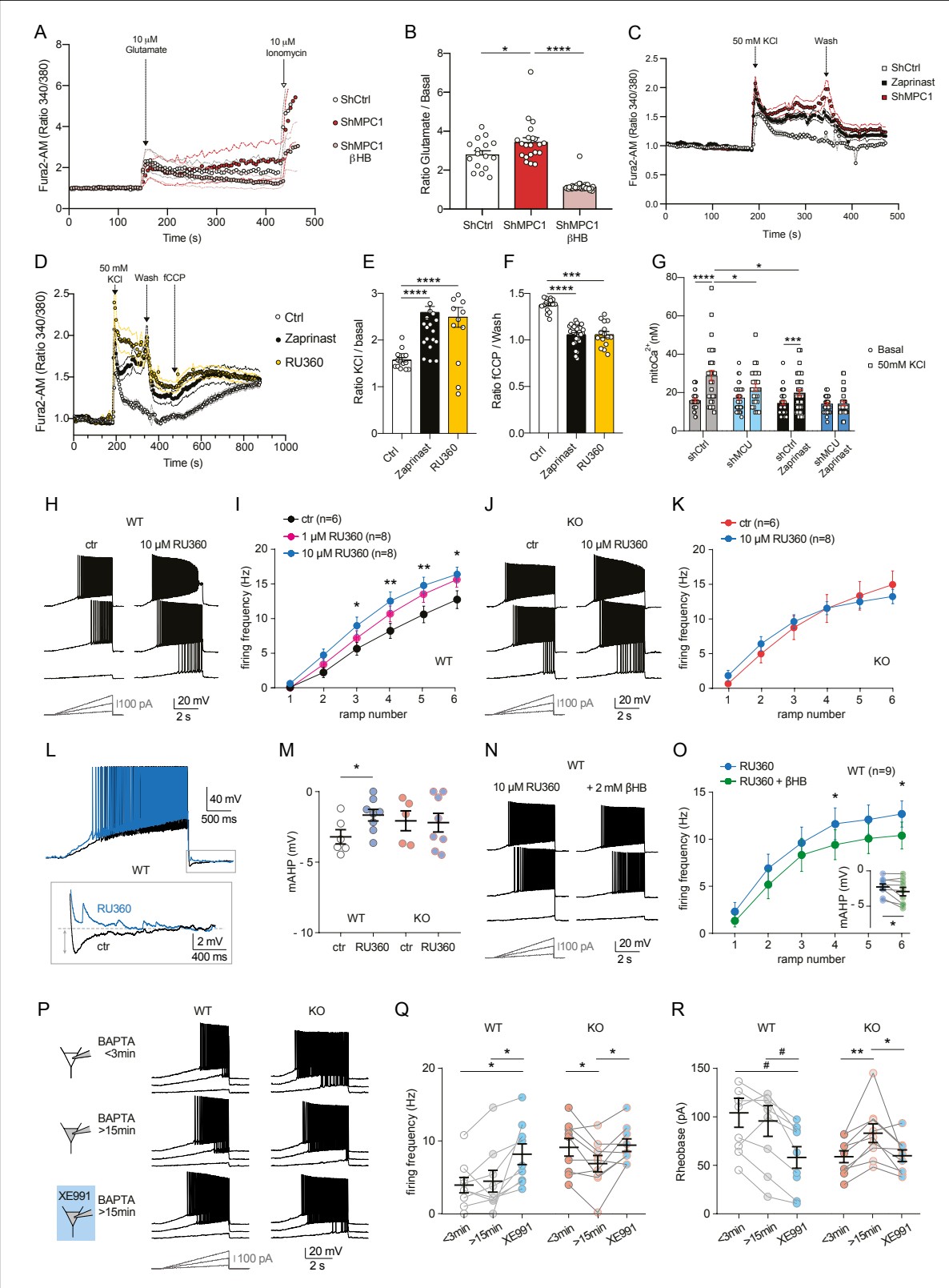

**Figure 6.** Defect in calcium homeostasis. (**A**) Mean fluorescence signal intensity of cortical neurons loaded with Fura2-AM stimulated with 10 μM glutamate (dashed black arrow) prior the addition of ionomycin (red arrow) to reveal the neuronal calcium stock. (**B**) Graph showing the quantifications of control neurons, MPC-depleted neurons and MPC-depleted neurons+βHB showing an elevated level of cytosolic calcium in MPC-deficient stimulated neurons measured by Fura2-AM. N > 15 neurons per condition from three independent experiments. One-way ANOVA + Holm Sidak's post-hoc test

*Figure 6 continued on next page*

*Figure 6 continued*

(shCtrl vs shMPC1 p = 0.0169, shMPC1 vs shMPC1+βHB p = 0.0001). (**C**) Mean fluorescence signal intensity of cortical neurons loaded with Fura2-AM before and after stimulation with 50 mM KCl (dashed black arrow) for 3 min. Neurons were then washed using a calcium-free medium. (**D**) Fluorescence signal intensity of control, MPC-deficient, and RU360-treated cortical neurons permeabilized with pluronic acid (0.02%). Neurons were loaded with Fura2-AM, stimulated with KCl 50 mM prior addition of fCCP to reveal the mitochondrial stocks of calcium. (**E, F**) Quantification of calcium increased upon depolarization (**E**) ratio of the fluorescence peak after adding KCl to the mean of the 10 first basal measurement) and the amount of mitochondrial calcium released by fCCP (**F**) ratio of the fluorescence peak after adding fCCP to the lowest point during wash in normal, MPC-deficient neurons and neurons + RU360. N > 13 neurons per condition from three independent experiments. One-way ANOVA + Holm Sidak's post-hoc test ((**E**) Ctrl vs Zaprinast p < 0.0001, Ctrl vs RU360 p < 0.0001; (**F**) Ctrl vs Zaprinast p = 0.0003, Ctrl vs RU360 p < 0.0001). (**G**) Mitochondrial calcium measurement of AAV-infected cortical neurons expressing mtaequorin. The MCU was inhibited by RNA interference (shMCU), whereas the MPC was inhibited using the MPC inhibitor Zaprinast (5 µM). The bioluminescence values were converged into calcium concentrations according to Bonora et al. Two-way ANOVA + Holm-Sidak post-hoc test (No significant difference could be observed between groups in. basal conditions. shCtrl basal vs shCtrl KCl, p < 0.0001, shMCU basal vs shMCU KCl, p = 0.067, shCtrl+ Zaprinast basal vs shCtrl+ Zaprinast KCl, p = 0.0008, shMCU+ Zaprinast basal vs shMCU+ Zaprinast KCl, p = 0.99, shCtrl KCl vs shMCU KCl, p = 0.012, shCtrl KCl vs shCtrl+ Zaprinast KCl, p = 0.011. (**H, I**). Example traces and F-I relationship in WT cells with standard intracellular solution and with a solution containing the MCU inhibitor RU360 (one or 10 µM), which increased neuronal firing (10 µM: Two-way ANOVA, $F_{(1, 72)}$ = 26.03, p < 0.0001). (**J, K**) Lack of RU360 (10 µM) effect on neuronal firing in KO cells (Two-way ANOVA, $F_{(1, 72)}$ = 0.03607, p = 0.8499). (**L**) Example traces of WT cell firing elicited by a current ramp (300 pA max amplitude, APs are trimmed) in control condition and with RU360, with expanded portion at the bottom indicating mAHP measurement. (**M**) Summary graph of mAHP values in control condition and with RU360, indicating significant reduction in WT (unpaired t test, t = 2.352, p = 0.0392). (**N, O**) Example traces and F-I relationship in WT cells infused with RU360 (10 µM) and subsequently exposed to βHB (2 mM, > 20 min exposure), which decreased neuronal firing (Two-way ANOVA, $F_{(1, 28)}$ = 17.69, p = 0.0001) and augmented mAHP (inset, paired t test, t = 2.336, p = 0.0477). (**P**) Scheme of whole-cell recordings with an intracellular solution containing the calcium chelator BAPTA (10 mM), with representative voltage responses in WT and KO neurons. Cell firing was compared between the first 3 min after whole-cell establishment, after 15 min of recordings, and after subsequent bath-application of XE991 (10 µM). (**Q**) Average firing frequency elicited by the 3rd ramp of current injections, indicating no altered intrinsic excitability in WT cells upon diffusion of BAPTA into the cytoplasm, but increased firing by subsequent bath application of XE991 (One-way ANOVA, $F_{(1.565, 12.52)}$ = 8.29, p = 0.0074, Holm-Sidak's multiple comparison *p < 0.05). In KO neurons, diffusion of BAPTA induced a significant decrease in cell firing, which was reversed by subsequent bath-application of XE991 (One-way ANOVA, $F_{(1.553, 12.42)}$ = 4.34, p = 0.0449, Holm-Sidak's multiple comparison *p < 0.05). (**R**) Changes in firing frequency were accompanied by a tendency toward decreased rheobase in WT (One-way ANOVA, $F_{(1.254, 10.03)}$ = 8.05, p = 0.014, Holm-Sidak's multiple comparison #p = 0.0512) and by significant changes in rheobase in KO neurons (One-way ANOVA, $F_{(1.5, 12)}$ = 7.974, p = 0.0093, Holm-Sidak's multiple comparison *p < 0.5, **p < 0.01).

The online version of this article includes the following figure supplement(s) for figure 6:

**Figure supplement 1.** Profiles of calcium imaging in cultured control and MPC1-deficient cortical neurons.

**Figure supplement 2.** Depletion of the mitochondrial calcium uniporter phenocopies MPC1-inhibition.

**Figure supplement 3.** Alterations in excitability of CA1 pyramidal cells from neuro-MPC1-KO mice are preserved under recording conditions that minimize cell dialysis and interference with intracellular ATP levels.

neurons compared to control (*Figure 6D,F*; *Figure 6—figure supplement 2B-D*). These results suggested a decreased storage of calcium in mitochondria from MPC- and MCU-deficient neurons. The mitochondrial aequorin probe was then used to test whether this was the consequence of reduced intake of calcium by mitochondria from these neurons, upon activation. As expected, we found that upon neuronal depolarization with KCl, the chemiluminescence of aequorin increased significantly in control and MPC and/or MCU deficient neurons (*Figure 6G*). However, in cells lacking either MPC or MCU activity the aequorin signal was much lower compared to control and no additivity was found when both MPC and MPU were inactivated together (*Figure 6G*). Taken together, these results show that the increased cytosolic calcium in MPC and/or MCU-deficient neurons results, at least in part, from decreased import of calcium into mitochondria. The low mitochondrial membrane potential in MPC-deficient neurons most likely explains this result.

To further test whether the increased cytosolic calcium resulting from dysfunctional mitochondria was responsible for the hyperexcitability of MPC1-deficient neurons, we performed electrophysiological recordings in CA1 pyramidal cells in the presence of RU360 into the patch pipette. In neuro-MPC1-WT cells, addition of 1 µM or 10 µM RU360 to the cell pipette caused an increase in cell firing (*Figure 6H,I*) while 10 µM RU360 had no effect in neuro-MPC1-KO cells (*Figure 6J,K*). Importantly, blockade of the MCU with RU360 in neuro-MPC1-WT cells was accompanied by a significant reduction in the mAHP (*Figure 6L,M*), indicating that calcium alterations induced by mitochondrial dysfunction may indeed affect M-type K⁺ channel activation. Finally, although βHB treatment did not significantly alter the firing of neuro-MPC1-WT cells in control conditions (*Figure 5E,F*), it reduced the excitability

of cells infused with 10 µM RU360 while slightly increasing mAHP (*Figure 6N,O*), consistent with the hypothesis that βHB normalizes the alteration in the M-type K + conductance.

To further substantiate the hypothesis that higher cytosolic calcium impairs M-type K$^+$ channel activity in neuro-MPC1-KO cells, we performed patch-clamp experiments using an intracellular solution containing the calcium chelator BAPTA (10 mM). We compared cell firing between the first minutes after the establishment of the whole-cell condition and 15 min thereafter, that is before and after complete diffusion of BAPTA in the cytoplasm (*Figure 6P–R*). Whereas intrinsic excitability was unaltered in neuro-MPC1-WT cells at these two timepoints, neuro-MPC1-KO cells displayed significantly lower firing rate and higher rheobase upon calcium chelation (*Figure 6Q,R*). Notably, unlike in BAPTA-free recordings (*Figure 5J,K,L*), subsequent bath application of XE991 (10 µM) increased intrinsic excitability not only in neuro-MPC1-WT cells, but also in neuro-MPC1-KO cells, indicating that calcium chelation made available a contribution of M-type K$^+$ channels that was otherwise impaired by the high intracellular calcium levels.

Altogether, our results show that MPC1-deficient neurons display a lower mitochondrial calcium buffering capacity which may explain the hypoactivity of the M-type K$^+$ channel and the intrinsic hyperexcitability of neurons.

## Discussion

Tight coupling between neuronal activity and energy metabolism is essential for normal brain function. Here, to assess the contribution of pyruvate metabolism in neuronal activity, we inactivated the MPC specifically in adult CamKIIα-expressing neurons in the mouse. As previously reported (*Divakaruni et al., 2017*; *Grenell et al., 2019*), we found that loss of the MPC led to decreased oxygen consumption and ATP production in glutamatergic neurons. Despite this, these mice appeared normal at rest and presented a normal behavioral repertoire (i.e. novelty exploration, sociability, stress coping), except for lower anxiety-like behaviors which are consistent with a higher glutamatergic tone (*Cordero et al., 2016*). Most strikingly, however, these mice developed severe seizures immediately following low-level administration of two pro-convulsant drugs, the GABA receptor antagonist pentylenetetrazole (PTZ), or the glutamate receptor agonist kainic acid.

The lack of an apparent phenotype in neuro-MPC1-KO mice at rest suggests that, up to a certain point, mitochondria can compensate for the deficit in mitochondrial pyruvate import by using other substrates to fuel the TCA cycle. It seems unlikely that such a compensatory mechanism would involve β-oxidation of fatty acids since these neurons do not express the enzymes necessary for this process (*Schönfeld and Reiser, 2013*). Furthermore, it is unlikely that the astrocyte-neuron-shuttle, which supplies astrocyte-derived lactate to neurons to boost OXPHOS (*Mächler et al., 2016*), can circumvent the loss of the MPC since all available data thus far indicate that lactate must first be converted into pyruvate by neuronal LDH in order to fuel the TCA cycle. It has been recently reported that inactivation of the MPC in cultured neurons, using the MPC small molecule inhibitor UK5099 (*Divakaruni et al., 2017*), or MPC1 gene knockout in the developing retina specifically (*Grenell et al., 2019*), resulted in a switch in mitochondrial substrate metabolism leading to increased reliance on glutamate (*Divakaruni et al., 2017*) or in ketones (*Grenell et al., 2019*) to fuel energetics and anaplerosis. Furthermore, inactivation of the MPC in cultured neurons also increased oxidation of the branched chain keto-acid catabolites of leucine, isoleucine, and valine (*Divakaruni et al., 2017*). Whether oxidation of these substrates can circumvent decreased pyruvate oxidation to allow MPC-deficient neurons to sustain normal function in neuro-MPC1-KO mice at rest remains to be confirmed.

Despite the lack of an obvious phenotype in resting mice, we found that, when challenged with the pro-convulsant molecules PTZ or kainic acid, the neuro-MPC1-KO mice were far more sensitive than WT animals and rapidly exhibited severe acute seizures. This suggests that the basal electrical activity of MPC1-deficient neurons may be continuously counterbalanced by inhibitory synapses, providing the normal resting phenotype described above. However, upon release of the 'brakes' exerted by the inhibitory system, the neuro-MPC1-KO neurons would become hyperactive, which would translate into the observed epileptic output. Consistent with our data, mice deficient in pyruvate dehydrogenase (PDH), the enzyme acting immediately downstream of the MPC, were found to display an epileptiform cortical activity accompanied by behaviorally observable seizures (*Jakkamsetti et al., 2019*). In this case, the epileptiform activity occurred in the context of reduced background cortical activation and, as suggested by the authors, the most likely explanation was that seizures resulted from

a combination of decreased activity of inhibitory neurons, mostly parvalbumin-expressing cortical neurons, with slightly overexcitable excitatory neurons. Similar to PDH-deficient neurons, we found that the MPC1-deficient neurons displayed higher input resistance and increased spike frequency after stimulation, a phenotype that we investigated further and found to be mediated by an impairment of the medium component of the after-hyperpolarization potential mediated by an M-type K + conductance.

$K^+$ efflux is the primary force behind the cellular repolarization that limits the spike after depolarization and thereby prevents neuronal hyperexcitability. One important class of $K^+$ channels that fulfills this task is the M-current ($I$M)-generating KCNQ channel family (also called Kv7 channels) (*Wang et al., 1998*). In hippocampal neurons the $I$M is mediated by the KCNQ2 and KCNQ3 channels (Kv7.1 and Kv7.2), which form hetero or homodimers. Loss of function of KCNQ2 or KCNQ3 causes epilepsy in humans and mice (*Biervert et al., 1998*; *Schroeder et al., 1998*; *Singh et al., 2008*; *Watanabe et al., 2000*). In support of the involvement of these channels in the intrinsic membrane hyperexcitability of MPC1-KO neurons, we found that their inhibition, using the small molecule XE991, did not change the electrical properties of KO neurons, while it made WT neurons more excitable. Our results suggest that KCNQ2/3 channels are closed in MPC1-deficient neurons, and that this could underlie their hyperexcitability.

Our observations point to dysregulation of calcium homeostasis as the cause for the silencing of these channels. High levels of cytosolic calcium have been reported to decrease KCNQ channel activity by detaching calmodulin from the channel or by inducing changes in the configuration of the calmodulin-KCNQ channel complex (*Alaimo and Villarroel, 2018*; *Kosenko and Hoshi, 2013*). In our study, we report a significant increase of cytosolic calcium levels in cultured depolarized MPC-deficient neurons due to an impaired capacity of mitochondria to import calcium. Accordingly, increasing cytosolic calcium levels in wild type neurons from acute hippocampal slices using the MCU inhibitor RU360 was sufficient to increase their firing properties, while RU360 had no significant effect on the excitability of the neuro-MPC1-KO neurons. Conversely, the intracellular calcium chelator BAPTA decreased intrinsic excitability in neuro-MPC1-KO neurons only and rendered them sensitive to XE991 blockade, suggesting that buffering the aberrant intracellular calcium levels was sufficient to reactivate the M-type $K^+$ conductance and normalize the excitability of MPC1 deficient neurons. Altogether, these results argue in favour of mitochondrial-related calcium homeostasis dysregulation as the reason for the silencing of KCNQ2/3 channels in neuro-MPC1-KO neurons.

The ketogenic diet has been reported to decrease seizures in patients with pharmacologically refractory epilepsy (*Carroll et al., 2019*) and we now report that the hyperexcitability of neuro-MPC1-KO mice fed with ketones is significantly reduced. Several hypotheses have been proposed to explain how ketone bodies could reduce neuron excitability, some involving direct pharmacological effects while others argue for an indirect role as metabolic fuels (*Yellen, 2008*). Beta-hydroxybutyrate (βHB) has been previously reported to display a direct pharmacological effect on the M-channel (*Manville et al., 2020*) and its protective action against proconvulsant drugs in neuro-MPC1-KO mice could be at least explained by this mechanism. On the other hand, acetoacetate, which is unlikely to impact directly the M-channel, was as efficient as βHB in reducing PTZ-induced seizures in neuro-MPC1-KO mice. This, together with the ability of βHB to restore oxygen consumption, ATP production, and mitochondrial membrane potential in cultured MPC-deficient neurons, suggests that one of the main protective actions of ketone bodies in neuro-MPC1-KO mice may be through provision of acetyl-CoA to the TCA cycle. However, additional mechanisms cannot be excluded. It has been previously reported that decreased glycolysis induced by ketones bodies can trigger activation of $K_{ATP}$ channels, thereby making neurons less excitable (*Giménez-Cassina et al., 2012*; *Yellen, 2008*). In our study, we show that MPC inhibition leads to increased aerobic glycolysis, which can be reversed by βHB. Whether $K_{ATP}$ channels are involved in the protection conferred by ketones in neuro-MPC1-KO mice requires further investigations.

In conclusion, using mice carrying an inducible deletion of the MPC specifically in excitatory neurons, we have shown that, despite impaired pyruvate-mediated oxygen consumption and ATP production, glutamatergic neurons can sustain high firing and trigger severe behaviorally observable seizures when the GABAergic network is inhibited. Furthermore, our data provide an explanation for the paradoxical hyperactivity of excitatory neurons resulting from OXPHOS deficits, which often

accompanies neuropathologies such as cerebral ischemia or diverse mitochondriopathies, and identify KCNQ channels as interesting therapeutic targets to prevent seizures occurring in these pathologies.

# Materials and methods

## Key resources table

| Reagent type (species) or resource | Designation | Source or reference | Identifiers | Additional information |
|---|---|---|---|---|
| Strain, strain background (Mouse, *M. musculus*) | *CamkIIa*$^{CreERT2}$ mice | Jackson laboratory | RRID: IMSR_JAX:012362 | |
| Strain, strain background (Mouse, *M. musculus*) | *Mpc1*$^{flox/flox}$ mice | gift from Professor Eric Taylor | NA | |
| Strain, strain background (Mouse, *M. musculus*) | Ai14 reporter mice | gift from Professor Ivan Rodriguez | NA | |
| Strain, strain background (Mouse, *M. musculus*) | *Gfap*$^{CreERT2}$ | gift from Professor Nicolas Toni | NA | |
| Strain, strain background (Mouse, *M. musculus*) | Neuro-MPC1-KO (*CamkIIa*$^{CreERT2}$ *MPC*$^{flox/flox}$ mice) | This article | NA | See materials and methods (p 18,19) |
| Strain, strain background (Mouse, *M. musculus*) | Astro-MPC1-KO (*Gfap*$^{CreERT2}$ *Mpc*$^{flox/flox}$ mice) | This article | NA | See materials and methods (p 18,19) |
| Cell line (*Homo sapiens*) | HEK293T | ATCC | RRID: CVCL_0045 | |
| Biological sample (*Mus musculus*) | Cultures of primary cortical neurons | This article, *Fauré et al., 2006* | | See materials and methods (p 20) |
| Biological sample (*Mus musculus*) | Organotypic cultures of hippocampus | This article, *Marissal et al., 2018* | | See Appendix 1 |
| Biological sample (*Mus musculus*) | Acute hippocampal slices | This article | NA | See materials and methods (p 19,20) |
| Recombinant DNA reagent | PLKO.1_shMCU | Sigma Aldrich (NM_001033259) | TRCN0000251263 | |
| Recombinant DNA reagent | PLKO.1_shMPC1 | Sigma Aldrich (NM_016098) | TRCN0000005485 TRCN0000005487 | |
| Recombinant DNA reagent | PLKO.1_shCTRL | Sigma Aldrich (SHC016) | | |
| Recombinant DNA reagent | AAV-GCaMP6s | University of Pennsylvania Vector Core | | |
| Recombinant DNA reagent | Ade-mtaequorin | Gift from Rosario Rizzuto lab, *Tosatto et al., 2016* | | |
| Antibody | anti-MPC1 (rabbit) | Sigma Aldrich (HPA045119) | RRID: AB_10960421 | 1/2000 (WB) 1/1000 (IF) 1/250(IHC) |
| Antibody | anti-MPC2 (Mouse) | Millipore (MABS1914) | MABS1914 | 1/500 (WB) |
| Antibody | anti-bIII-tubulin (Mouse) | Biolegend (801201) | RRID:AB_2313773 | 1/5000 (WB) 1/1000 (IF) |
| Antibody | anti-MCU (Rabbit) | Sigma Aldrich (HPA 016480) | RRID:AB_2071893 | 1/5000 (WB) |
| Antibody | anti-HSP70 (Mouse) | Millipore (MABS1955) | MABS1955 | 1/1000 (WB) |
| Antibody | Anti-TOMM20 (Rabbit) | Abcam (Ab186735) | RRID:AB_2889972 | 1/2000 (IF) |
| Antibody | anti-synaptophysin (Mouse) | Abcam (ab8049) | RRID:AB_2198854 | 1/2000 (WB) |
| Antibody | anti-Tyrosine Hydroxylase (Rabbit) | Millipore (AB152) | RRID:AB_390204 | 1/2000 (WB) |
| Antibody | anti-CamKIIa (Goat) | Abcam (ab87597) | RRID:AB_2040677 | 1/5000 (WB) |
| Antibody | anti-GFAP (Mouse) | Sigma Aldrich (G3893) | RRID:AB_477010 | 1/5000 (WB) |
| Antibody | anti-VDAC (Goat) | Santa Cruz (sc-8829) | RRID:AB_2214801 | 1/500 (WB) |
| Antibody | anti-bActin (Mouse) | Sigma Aldrich (a3854) | RRID:AB_262011 | 1/50000 (WB) |
| Antibody | goat anti-mouse Alexa Fluor488 | Life technologies (A32723) | RRID:AB_2633275 | 1/2000 |
| Antibody | goat anti-rabbit Alexa Fluor594 | Life technologies (A11037) | RRID:AB_2534095 | 1/2000 |

*Continued on next page*

*Continued*

| Reagent type (species) or resource | Designation | Source or reference | Identifiers | Additional information |
|---|---|---|---|---|
| Antibody | anti-IgG-Mouse-HRP | Dako (P0447) | RRID:AB_2617137 | 1/10000 |
| Antibody | anti-IgG-Rabbit-HRP | Dako (P0217) | RRID:AB_2728719 | 1/10000 |
| Antibody | anti-IgG-Goat-HRP | Santa Cruz (sc-2304) | RRID:AB_641158 | 1/10000 |
| Commercial assay or kit | CellTiter Glo | Promega | G9241 | |
| Commercial assay or kit | DeadEnd Colorimetric TUNEL System | Promega | G7130 | |
| Commercial assay or kit | GlucoMen Lx Plus kit | Menarini diagnostics | | |
| Other | Ketogenic diet | Provimi Kliba AG | XL75:XP10 | |
| Chemical compound, drug | FuraFF | Thermo Fisher Scientific | F14181 | |
| Chemical compound, drug | Fura2-AM | Thermo Fisher Scientific | F1221 | |
| Chemical compound, drug | Fluo4-AM | Thermo Fisher Scientific | F14201 | |
| Chemical compound, drug | 2-NDBG (2-(N-(7-Nitrobenz-2-oxa-1,3-diazol-4-yl)Amino)–2-Deoxyglucose) | Thermo Fisher Scientific | N13195 | |
| Chemical compound, drug | Pentylenetetrazol (PTZ) | Sigma Aldrich | P6500 | |
| Chemical compound, drug | Kainic Acid | Sigma Aldrich | K0250 | |
| Chemical compound, drug | Tamoxifen | Sigma Aldrich | 85,256 | |
| Software, algorithm | Prism7 | Graphpad version 7.0 a, April 2, 2016. | RRID:SCR_002798 | |
| Other | Digidata1550A digitizer | Molecular Devices | | |
| Other | Digital Lynx SX | Neuralynx, USA | | |
| Other | Seahorse XF 24 extracellular flux analyzer | Seahorse Biosciences | | |
| Other | Cytation 3TM | Biotek Instrument Inc | | |
| Other | IX71 Olympus microscope | Olympus | | |
| Other | Confocal LSM780 microscope | Zeiss | | |

## Study design

Data sources from mice included in vivo (behavioral tests, pro-convulsant drug injections, electroencephalogram), brain slice recordings of neuronal activity and electrophysiology, isolation of synaptosomes and primary culture of cortical neurons. For mouse experiments, pilot data from three or four samples per group provided an estimate of SD and effect magnitude, which, together with a power of 0.8 and $P < 0.05$, guided sample sizes using the G*power software (G*power version 3.1.9.6.). MPC1-WT and MPC1-KO mice from the same litter were randomly selected for experiments. Replicates and statistical tests are cited with each result. All procedures were approved by the Institutional Animal Care and Use Committee of the University of Geneva and with permission of the Geneva cantonal authorities. Data analysis was blind and performed concurrently on control and experimental data with the same parameters. No data, including outlier values, were excluded.

## Mice

The *CamkIIa^CreERT2* mouse was obtained from Jackson (stock number 012362). The *Mpc1^flox/flox* mouse was a gift from professor Eric Taylor (University of Iowa). The Ai14 reporter mouse was a gift from professor Ivan Rodriguez (University of Geneva). The *Gfap^CreERT2* mouse was a gift from professor Nicolas Toni (University of Lausanne) (*Gebara et al., 2016*). By using the Cre driver lines, we generated two different cell-type specific *Mpc1* KO mice: *CamkIIa^CreERT2-Mpc1^flox/flox* mice (here called neuro-MPC1-KO) in which *Mpc1* was knocked out specifically in excitatory glutamatergic neurons; and *Gfap^CreERT2-Mpc1^flox/flox* mice in which *Mpc1* is knockout specifically is astrocytes (here called

astro-MPC1-KO). In all experiments age-matched wild type controls were used and are referred to in the text as neuro-MPC1-WT (*CamKIIa*$^{CreERT2}$-*Mpc1*$^{flox/flox}$) and astro-MPC1-WT mice (*Gfap*$^{CreERT2}$-*Mpc1*$^{flox/flox}$). The neuro-MPC1-KO and astro-MPC1-KO phenotypes were tamoxifen-inducible. In order to induce MPC1 deletion, the mice were injected intraperitoneally (ip) for five consecutive days with 100 µl of 10 mg/ml tamoxifen (Sigma, 85256) in sunflower oil. The mice were considered to be MPC1-KO from 1 week after the final injection. All experiments were carried out in accordance with the Institutional Animal Care and Use Committee of the University of Geneva and with permission of the Geneva cantonal authorities (Authorization numbers GE/42/17, GE/70/15, GE/123/16, GE/86/16, GE/77/18, GE/205/17) and of the Veterinary Office Committee for Animal Experimentation of Canton Vaud (Authorization number VD3081).

## Pentylenetetrazol (PTZ)-induced convulsion protocol

We used the PTZ kindling model of epilepsy as described in *Dhir, 2012*. Briefly, this test entails chronic intraperitoneal (ip) injection of 35 mg/kg PTZ (Sigma, P6500), which is a sub-convulsant dose for WT mice, every 2 days for 2 weeks, and after each PTZ injection, the mice were scored according to their clinical symptoms, as described previously (*Dhir, 2012*; *Mishra et al., 2018*). After each PTZ injection, the animals were gently placed in isolated transparent plexiglass cages and their behavior was observed to assign a seizure score based on the following criteria: stage 1: sudden behavioral arrest and/or motionless staring; stage 2: jerks; stage 3: Straub's tail (rigid tail being held perpendicularly to the surface of the body); stage 4: partial clonus in a sitting position; stage 5: generalized clonus; stage 6: convulsions including clonic and/or tonic–clonic seizures while lying on the side and/or wild jumping (convulsive status epilepticus). Mice were scored over a period of 30 min and the tests were performed in semi-blind mode (carried out by 2 experimenters of which only one knew the genotype). After the PTZ test, mice were immediately sacrificed in a CO$_2$ chamber. The seizure severity score was calculated by taking the sum of the behavior and seizure patterns for all animals in a group and dividing by the number of animals present in the group.

## Electroencephalogram (EEG)

Surface EEGs were recorded in head-fixed, awake animals with 32 stainless steel electrodes (500 µm Ø) covering the entire skull surface as described previously (*Mégevand et al., 2008*; *Sheybani et al., 2018*). Briefly, a head-post was placed under isoflurane anaesthesia allowing head-fixation. Recording sessions took place after a period of 4 days of head-fixation training to allow acclimatization of the animals to the experimental setup. PTZ was injected ip at the beginning of the session. Electrophysiological differential recordings were acquired with a Digital Lynx SX (Neuralynx, USA) at a sampling rate of 4 kHz and with a 2 kHz low-pass. The ground electrode was placed above the nasal bone and the reference electrode was placed on the midline between parietal bones (channel 31, *Figure 3C*). All signals were calculated against the average reference offline.

## Patch-clamp electrophysiology

Tamoxifen-treated *CamKIIa*$^{CreERT2}$-*Mpc1*$^{flox/flox}$ mice and wild-type littermates (6–10 weeks-old) were anaesthetized with isoflurane and decapitated, and the brain was quickly removed and placed in oxygenated (95% O$_2$ / 5% CO$_2$) ice-cold N-Methyl-D-glucamine (NMDG)-based medium, containing (in mM): 110 NMDG, 2.5 KCl, 1.2 NaH$_2$PO$_4$, 30 NaHCO$_3$, 20 HEPES, 10 MgCl$_2$, 0.5 CaCl$_2$, 25 glucose, 5 L(+)-ascorbic acid, 2 thiourea, 3 Na-pyruvate (titrated to pH 7.2–7.3 with HCl). Acute hippocampal transverse slices (350 µm thick) were cut using a vibrating tissue slicer (Campden Instruments). Slices recovered for 1 hr at 35 °C and subsequently at room temperature in a storage solution containing (in mM): 92 NaCl, 2.5 KCl, 1.2 NaH$_2$PO$_4$, 30 NaHCO$_3$, 20 HEPES, 2 MgCl$_2$, 2 CaCl$_2$, 25 glucose, 5 L(+)-ascorbic acid, 2 thiourea, 3 Na-pyruvate (titrated to pH 7.2–7.3 with NaOH). In the recording chamber, slices were superfused with oxygenated standard artificial cerebrospinal fluid (aCSF) containing (in mM): 130 NaCl, 25 NaHCO$_3$, 2.5 KCl, 1.25 NaH$_2$PO$_4$, 1.2 MgCl$_2$, 2 CaCl$_2$, 18 glucose, 1.7 L(+)-ascorbic acid.

Whole-cell patch clamp recordings were performed at nearly physiological temperature (30°C–32°C), with borosilicate pipettes (3–4 MΩ) filled with (in mM): 130 KGluconate, 10 KCl, 10 HEPES, 10 phosphocreatine, 0.2 EGTA, 4 Mg-ATP, 0.2 Na-GTP (290–300 mOsm, pH 7.2–7.3). Whole-cell patch clamp recordings were performed at nearly physiological temperature (30°C–32°C), with borosilicate

pipettes (3–4 MΩ) filled with (in mM): 130 KGluconate, 10 KCl, 10 HEPES, 10 phosphocreatine, 0.2 EGTA, 4 Mg-ATP, 0.2 Na-GTP (290–300 mOsm, pH 7.2–7.3).

For recordings with intracellular calcium chelation, pipettes (3–4 MΩ) were filled with (in mM): 100 KGluconate, 10 KCl, 10 HEPES, 10 phosphocreatine, 10 1,2-Bis(2-aminophenoxy)ethane-N,N,N',N'-tetraacetic acid (BAPTA), 4 Mg-ATP, 0.2 Na-GTP (290–300 mOsm, pH 7.2–7.3). BAPTA was prepared in a 20 mM stock solution titrated with 50 mM KOH.

A control experimental series was conducted with narrow pipettes tips (9–10 MΩ) filled with (in mM): 130 KGluconate, 5 KCl, 10 HEPES, 5 Sucrose (275–280 mOsm, pH 7.2–7.3), in order to delay intracellular dialysis (*Jakkamsetti et al., 2019*) and minimize interference with intracellular ATP and $Ca^{2+}$ levels. In this series, neuronal firing was measured within the first 1.5 min after whole-cell establishment (*Figure 6—figure supplement 3*).

To elicit neuronal firing, cells were held at –60 mV with direct current injections, and somatic current injections of increasing amplitude were provided using ramps of 5 s (six ramps with final amplitude ranging from 50 pA to 300 pA) or squared pulses of 2 s (25 pA delta increase, max amplitude 200 pA). Input resistance (Ri) was assessed by the passive current response to a –10 mV hyperpolarizing step while cells were held at –60 mV. In control condition, resting membrane potential (Vrmp) and neuronal firing were measured within the first 5 min from the establishment of the whole-cell condition. The rheobase and the firing threshold were measured as the level of current and voltage, respectively, that induced the first action potential in the ramp protocol. The effect of β-hydroxybutyrate (2 mM) was assessed after >20 min perfusion, and compared to cell firing prior to perfusion.

Signals were acquired through a Digidata1550A digitizer, amplified through a Multiclamp 700B amplifier, sampled at 20 kHz and filtered at 10 kHz using Clampex10 (Molecular Devices).

## Cell culture and lentiviral transduction

Wild type pregnant mice were decapitated and E18 embryos were collected in HBSS medium. Primary cultures of cortical neurons were prepared as described previously (*Fauré et al., 2006*). Briefly, cortices were dissected from E18 mouse embryos in HBSS and treated with 0.25% trypsin-1mM EDTA for 15 min at 37 °C. Tissues were washed, transferred to DMEM seeding medium (DMEM, 10% horse serum, 0.5 mM L-glutamine) and dissociated by 7–8 cycles of aspiration and ejection through a micropipette tip. Neurons were seeded at 250,000 neurons per $cm^2$ on coverslips coated with 50 µg/ml poly-D-lysine. After 3 hrs, the seeding medium was replaced by serum-free neuronal culture medium (Neurobasal medium, 2% B27 supplement, penicillin/streptomycin and 0.5 mM L-glutamine). For MCU downregulation 7 DIV neurons were treated with lentiviral particles containing shRNA targeting MCU (NM_001033259/TRCN0000251263; Sigma Aldrich) (*Qiu et al., 2013*) for a further 7–8 days. For MPC1 downregulation 7 DIV neurons were treated with lentiviral particles containing shRNA targeting MPC1 for a further 7–8 days. Briefly, to prepare viral particles, Hek293T cells were transfected with packaging and envelope expressing plasmids together with PLKO.1-shRNA control (SHC016, SIGMA) or targeting MPC1 with the following sequences: ShMPC1_1:CCGGGCTGCCTTACAAGTATTAAATCTCGAGATTTAATACTTGTAAGGCAGCTTTTT; ShMPC1_2:CCGGGCTGCCATCAATGATATGAAACTCGAGTTTCATATCATTGATGGCAGCTTTTT. After 72 hours the culture supernatant was collected, ultracentrifuged at 100,000 g for 2 hr.

## Determination of oxygen consumption rate (OCR) and extracellular acidification rate (ECAR)

Measurement of oxygen consumption was performed using a Seahorse XF 24 extracellular flux analyzer (Seahorse Biosciences). A total of 80,000 cells were seeded in XF24 cell culture microplates and grown for 16 days. Measurement of basal and stimulation-dependent oxygen consumption was carried out at 37°C in aCSF (140 mM NaCl, 5 mM KCl, 1.2 mM $KH_2PO_4$, 1.3 mM $MgCl_2$, 1.8 mM $CaCl_2$, 5 mM Glucose, and 15 mM Hepes, pH 7.4). Cells were infected with control shRNA or shMPC1 as decribed above or treated with MPC1 inhibitors Zaprinast (*Du et al., 2013*), Rosiglitazone (*Divakaruni et al., 2013*), and UK5099 (*Halestrap, 1975*) at 5, 5, and 1 µM, respectively. Cells were treated as indicated in the figure legends for 30 min before performing the assay. Basal oxygen consumption was measured before injection. At the times indicated, the following compounds were injected: oligomycin (1 µM), fCCP (4 µM), Rotenone/Antimycin A (1 µM). Each measurement loop consisted of 30 s mixing, 2 min incubation, and 3 min measurement of oxygen consumption.

Determination of the extracellular acidification was carried out under the same conditions but in the absence of HEPES. The basal acidification rate was measured before injection. At the times indicated, the following compounds were injected: oligomycin (1 μM), 2-deoxyglucose (5 mM). Each measurement loop consisted of 2 min mixing, 2 min incubation, and 3 min measurement of oxygen consumption.

## ATP measurements

ATP measurements were performed on 14–17 DIV neurons, infected with control or MPC1 shRNA as described above, or treated 30 min prior to performing the assay with MPC inhibitors. Neurons were washed and scraped in PBS. Neurons were centrigugated at 1000 rpm for 5 min and resuspended in 100 mL of CellTiter Glo reagent and agigated for 2 min to allow cell lysis. After 10 min incubation, luminescence was recorded.

## Calcium imaging

E18.5 primary cortical neurons were isolated and seeded onto 35mm Fluorodishes or 96 well plate. Neurons were treated with control or MCU shRNAs at 7DIV and used for calcium imaging at 14–17 DIV.

### Cytosolic calcium

Neurons were loaded with 5 μM FuraFF, Fura2-AM or Fluo4-AM (F14181, F1221, and F14201, Thermo Fisher Scientific) in recording buffer (150 mM NaCl, 4.25 mM KCl, 4 mM NaHCO$_3$, 1.25 mM NaH$_2$PO$_4$, 1.2 mM CaCl$_2$, 10 mM D-glucose, and 10 mM HEPES at pH 7.4) with 0.02% pluronic acid, at 37 °C and 5% CO$_2$ for 30 min.

For FuraFF and Fura2, cells were washed and imaged in recording buffer using a custom-made imaging widefield system built on an IX71 Olympus microscope equipped with a 20×water objective. A Xenon arc lamp with a monochromator was used for excitation, exciting FuraFF or Fura2 fluorescence alternately at 340 nm ± 20 nm and 380 nm ± 20 nm and collecting emitted light through a dichroic T510lpxru or a 79003-ET Fura2/TRITC (Chroma), and a band-pass filter 535/30 nm. Neurons were stimulated using 50 mM KCl or 10 μM glutamate (G1626, Sigma) and 4 μM fCCP or 10 μM Ionomycin was added at the end of each time course experiment. Images were acquired using a Zyla CMOS camera (Andor) every 2–5 s. The images were then analysed using ImageJ. Briefly, Regions of Interest (ROIs) were selected and average fluorescence intensity was measured for each channel including the background fluorescence. After subtracting the background fluorescence, the ratio between 380 and 340nm was calculated and plotted as cytosolic [Ca$^{2+}$] levels upon stimulation. The Mean amplitude was calculated for each cell using Graphpad Prism.

For Fluo4-AM, cells were washed and imaged using a fluorescent plate reader (Cytation 3TM, Biotek Instrument Inc) with the following parameters: Excitation/Emission: 485/515, Gain:115. Fluo4-AM fluorescence was recorded for 3 min prior and 5min after neuron depolarization with injection of 50mM KCl. Then fCCP (4μM) was added to unload mitochondria from their calcium content.

### Mitochondrial calcium

For the quantification of mitochondrial calcium, neurons were infected with adenovirus allowing the expression of the mtaequorin wt (*Bonora et al., 2013*) 48 hr prior to imaging. Neurons were incubated with 5 μM coelenterazine for 3 hr in aCSF solution at RT. Luminescence emission was then recorded using a luminescent plate reader (Cytation 3TM, Biotek Instrument Inc) for 5 min prior to and after neuronal depolarization (50 mM KCl). The experiment was terminated by lysing the neurons with 100 μM digitonin in a hypotonic Ca$^{2+}$-rich solution (10 mM CaCl$_2$), thus discharging the remaining aequorin pool. The light signal was collected and calibrated into [Ca$^{2+}$] values as previously described (*Bonora et al., 2013*).

## Statistical analysis

The comparison of two groups was performed using a two-sided Student's t-test or its non parametric correspondent, the Mann-Whitney test, if normality was not granted either because not checked (n < 10) or because rejected (D'Agostino and Pearson test). The comparisons of more than two groups were made using one or two ways ANOVAs followed by post-hoc tests, described in the figure

legends, to identify all the significant group differences. N indicates independent biological replicates from distinct samples. Data are all represented as scatter or aligned dot plot with centre line as mean, except for western blot quantifications, which are represented as histogram bars. The graphs with error bars indicate 1 SEM (+/-) and the significance level is denoted as usual (#$p < 0.1$, *$p < 0.05$, **$p < 0.01$, ***$p < 0.001$). All the statistical analyses were performed using Prism7 (Graphpad version 7.0 a, April 2, 2016). version 7.0 a, April 2, 2016.

## Acknowledgements

We would like to express our sincere thanks to Drs. Nika Danial (Harvard Medical School), Timothy Ryan (Weill Cornell Medicine), Garry Yellen (Harvard Medical School) and all members of the Martinou lab for helpful scientific discussion during the course of this work. We are also thankful to Denis Vecellio Reane from Rosario Rizzuto lab for sharing the Adenovirus-mtaequorin construct and for advices on mitochondrial calcium imaging. We also wish to extend our special thanks to Dr. Fabien Lanté and Prof. Anita Lüthi for advices on the electrophysiology experiments, to Dr. Kinsey Maundrell for help in reviewing the manuscript, to Professor Nicolas Toni (University of Lausanne) who kindly provided us with the Gfap$^{CreERT2}$ mouse, and to Professor Ivan Rodriguez (University of Geneva) who kindly provided us with the Ai14 reporter mouse.

## Additional information

### Funding

| Funder | Grant reference number | Author |
|---|---|---|
| Ministerio de Economía, Industria y Competitividad, Gobierno de España | BFU2017-84490-P | Pablo Mendez |
| Swiss National Science Foundation | 31003A_179421/1 /1 | Jean-Claude Martinou |
| Kristian Gerhard Jebsen Foundation | | Carmen Sandi Jean-Claude Martinou |

The funders had no role in study design, data collection and interpretation, or the decision to submit the work for publication.

### Author contributions

Andres De La Rossa, Data curation, Data curation, Formal analysis, Methodology, Writing – original draft; Marine H Laporte, Data curation, Data curation, Formal analysis, Validation, Writing – original draft, Writing – review and editing; Simone Astori, Data curation, Formal analysis, Writing – review and editing; Thomas Marissal, Eva Ramos-Fernández, Pablo Mendez, Abbas Khani, Data curation, Formal analysis; Sylvie Montessuit, Data curation, Methodology; Preethi Sheshadri, Formal analysis, Data curation; Charles Quairiaux, Alan Carleton, Michael R Duchen, Supervision; Eric B Taylor, Jared Rutter, Resources; José Manuel Nunes, Software, Validation; Carmen Sandi, Supervision, Writing – original draft, Writing – review and editing; Jean-Claude Martinou, Data curation, Funding acquisition, Supervision, Validation, Writing – original draft, Writing – review and editing

### Author ORCIDs

Marine H Laporte http://orcid.org/0000-0002-7856-6763
Simone Astori http://orcid.org/0000-0001-7698-8332
Eva Ramos-Fernández http://orcid.org/0000-0002-3771-2189
Pablo Mendez http://orcid.org/0000-0001-9862-6818
Abbas Khani http://orcid.org/0000-0001-7324-4303
Charles Quairiaux http://orcid.org/0000-0003-3770-8232
Eric B Taylor http://orcid.org/0000-0003-4549-6567
Jared Rutter http://orcid.org/0000-0002-2710-9765
José Manuel Nunes http://orcid.org/0000-0001-7010-1382
Alan Carleton http://orcid.org/0000-0001-5633-9159

Carmen Sandi http://orcid.org/0000-0001-7713-8321
Jean-Claude Martinou http://orcid.org/0000-0002-9847-2051

### Ethics

All experiments were carried out in accordance with the Institutional Animal Care and Use Committee of the University of Geneva and with permission of the Geneva cantonal authorities (Authorization numbers GE/42/17, GE/70/15, GE/123/16, GE/86/16, GE/77/18, GE/205/17) and of the Veterinary Office Committee for Animal Experimentation of Canton Vaud (Authorization number VD3081).

### Decision letter and Author response

Decision letter https://doi.org/10.7554/eLife.72595.sa1
Author response https://doi.org/10.7554/eLife.72595.sa2

## Additional files

### Supplementary files

• Source data 1. Uncropped western blot corresponding to *Figure 1—figure supplement 1A*, *Figure 1—figure supplement 1B*, *Figure 2C*, *Figure 3—figure supplement 1E*, *Figure 6—figure supplement 2A*.

• Transparent reporting form

### Data availability

The data that are supporting the findings of this study are included within the article as source data files.

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

# Appendix 1.

## Food and diet

Standard diet is composed of 4.9% fat, 24% crude proteins, 29% starch (Provimi Kliba AG, standard diet, 3800). The ketogenic diet (KD) is composed of 74.4% animal fat, 9.9% crude protein, 0.7% starch (Provimi Kliba AG, Ketogenic diet XL75:XP10).

## Glycaemia and ketonemia measurements

Glycaemia and ketonemia were measured in blood using the GlucoMen Lx Plus kit (Menarini diagnostics) which has a sensitivity for ketones, mainly $\beta$HB, from 0.1 mM to 8 mM. If not stated otherwise, measurements were performed using one drop of blood from the tail-veins of mice fed ad-libitum.

## Behavioural tests

In each behavioral test, 6–10 adult animals per group were analyzed. They were tested in two tests to evaluate their anxiety-like responses: the elevated plus maze (EPM) and open field (OF), followed by a sociability test and a test for stress coping behaviors, the forced swim test (FST), all described below. Testing took place with at least 5 days between paradigms. The **EPM** is a maze composed of two open and two closed arms (30 × 5 × 14 cm) separated by a central platform (5 × 5 cm). The mouse was introduced in the center of the maze facing the wall of the closed arm and allowed to explore freely for 5 min. The intensity of the lights was maintained at 12 lux on the open arms, 10 lux in the center and 4–3 lux in the closed. After each trial, the maze was cleaned with 5%–7% ethanol and dried. Video tracking of the animal's location was performed by a camera placed above the arena. The percentage of time spent in the open arms, as a measure of anxiety, was calculated with Ethovision (Noldus SA) tracking system. The **OF** is an arena (50 × 50 cm) in which the experimental mouse is placed in one corner facing the wall and left to freely explore for 10 min. Lighting was maintained at 7 lux on the center. After each trial, the arena was cleaned with 5%–7% ethanol and dried. Video tracking of the animal was recorded by a camera fixed above the arena, and images were processed using the Ethovision tracking system. The percentage of time spent in the center of the open field was taken as an indicator of anxiety. In the **Social preference (SP) test**, the preference for either a social stimulus or an inanimate object is tested. The arena consists of a rectangular, three-chambered box with a center (20 × 35 x 35 cm) and two side compartments (30 × 35 × 35 cm) with sawdust covering the floor. Retractable doorways are located between the compartments to allow or prevent access to the side chambers. In both side chambers, an object (black dummy mouse) or a juvenile mouse C57BL/6 J (28 ± 2 days) are located in a wire cage (10.16 cm bottom diameter, 11 cm high, bars spaced 0.5 cm apart). Light conditions are kept at 10 lux in the center. Two consecutive days prior to testing, experimental animals were habituated to the apparatus for 10 min. Juvenile mice were also habituated to the wire cages for 30 min. In the test day, each experimental mouse was located in the center compartment with both doors closed for 5 min. The doors were then opened and the mouse was allowed to explore the arena for 10 min. Behavior was video-recorded and the percentage of time each mouse spent sniffing either the object or the juvenile scored with the Observer (Noldus SA) software. The **Forced swim test (FST)** involves placing each mouse in in a beaker containing 3 L of tempered water (24°C ± 1°C) for 6 min. Behavior was recorded with a video camera and the immobility time spent during the last 4 min quantified with the Ethovision (Noldus S.A.) tracking system.

## Echo magnetic resonance (EchoMRI)

The body composition was measured with an echo magnetic resonance (Echo Medical System). Briefly, the mouse was weighed before scanning. To calibrate the device, the mouse weight was introduced and the animal was placed on the holder inside the scanner. The % of lean mass was calculated considering the total body weight of each animal.

## lectrophysiological field recordings

Acute hippocampal slices were prepared as described in the main Materials and Methods section. Field recordings were conducted at nearly physiological temperature (30°C–32°C) in the presence of the GABA$_A$R blocker picrotoxin (0.1 mM). Field excitatory postsynaptic potentials (fEPSPs) were acquired in CA1 stratum radiatum through a borosilicate pipette filled with aCSF while stimulating Schaffer collaterals (0.1ms duration) with a tungsten concentric microelectrode (World Precision Instruments). Input-output curves of fEPSPs were constructed by displaying the fEPSP initial slope

as a function of the amplitude of the presynaptic volley (FV), which was increased by augmenting the stimulus intensity (50–500 µA). Two consequent stimuli were provided with an interval of 50ms. Paired-pulse ratio was calculated by dividing the initial slope of the second response by the initial slope of the first response.

Signals were acquired through a Digidata1550A digitizer, amplified through a Multiclamp 700B amplifier sampled at 4 kHz and filtered at 1 kHz for field recordings, using Clampex10 (Molecular Devices).

## Calcium imaging of the network dynamics

Hippocampal slice cultures prepared as previously described (*Marissal et al., 2018*; *Stoppini et al., 1991*) were infected with virus to enable expression of the genetically encoded calcium sensor GCaMP6s under the control of the human synapsin promoter (University of Pennsylvania Vector Core). Slices were immersed in an artificial cerebro-spinal fluid containing 124 mM NaCl, 1.6 mM KCl, 1.2 mM $KH_2PO_4$, 1.3 mM $MgCl_2$, 2.0 mM $CaCl_2$ 10 mM Glucose, and 2.0 mM ascorbic acid, pH 7.4, at room temperature (22 °C) and continuously perfused and oxygenated using a peristaltic pump. Calcium transients were recorded using a Nipkow-type spinning disk confocal microscope (Olympus) coupled with single-photon laser (excitation wavelength 488). Images were acquired through a CCD camera (Visitron Systems Evolve). Slices were imaged using a 10 × 0.30 NA objective (Olympus) at 8.9 Hz frame rate (i.e. 112ms per frame). Typically, imaging covered a field of 420 × 420 µm containing ~200 individual neurons in the CA1 area. For treatment with pentylenetetrazol (PTZ), slices were pre-incubated with 2 mM drug for 15 minutes, and, slices were imaged after addition of 50 µM carbachol (CCh), which induces the generation of activity patterns mimicking those recorded in vivo (*Fisahn et al., 1998*). All drugs remained present throughout the recordings.

Calcium data analyses were performed using the Matlab-based 'Caltracer3beta' software (Columbia University), and custom-made scripts as previously described (*Marissal et al., 2018*). This enabled (a) tracing of Regions Of Interests (ROIs) corresponding to the identified GCaMP6s-expressing neuronal soma, and (b) the calculation of the averaged fluorescence signal from each ROIs as a function of time. In order to distinguish the neuronal calcium activity from background activity (i.e. from nearby fibres), the fluorescence within a halo of the pixels surrounding the ROI was subtracted from the fluorescence signal recorded in the ROI. From the resulting signal, the onset and offsets of the calcium events were identified. Onsets were automatically detected when fluorescence in a given slice exceeded a threshold value (based on the background noise) for a minimum duration of 1 s (based on the kinetics of GCaMP6s), and offsets were defined as the half-decay time of the event. Onset and offsets were manually corrected after automatic detection, and used to estimate (1) the frequency, the amplitude, and the duration of the calcium events, and (2) the occurrence of co-activation (e.g. neuronal synchronization) exceeding random chance.

## Isolation of synaptic membranes

Isolation of synaptic membranes (hereafter referred to as synaptosomes) from P70 neuro-MPC1-WT and neuro-MPC1-KO mice was performed according to *Chassefeyre et al., 2015* with minor modifications. Briefly, cortices from one mouse were homogenized in 2 ml ice-cold isotonic lysis buffer (0.32 M sucrose, 4 mM Hepes pH 7.4, protease inhibitors) using a teflon-glass-homogenizer (10 strokes). The homogenate was centrifuged at 1,000 x g for 10 min to remove nuclei. The post-nuclear supernatant (S1) was spun down at 4 °C for 20 min at 13,800 x g, to yield crude synaptosomes (pellet P2) and crude cytosol (supernatant S2). The P2 fraction was homogenized in lysis buffer and layered onto discontinuous sucrose density gradients consisting of 3 ml each of 0.8 M, 1.0 M, 1.2 M sucrose in 4 mM Hepes, pH 7,4 containing protease inhibitors. The gradients were centrifuged at 82,500 x g in a Beckman SW41 rotor for 120 min and the synaptosomes were collected at the 1.0–1.2 M interface, resuspended in 5 vol. lysis buffer and spun down for 20 min at 150,000 g. The pelleted synaptosomes were resuspended in the appropriate buffer for either western blotting and oxygen consumption rate as described below under separate subheadings.

## Determination of oxygen consumption rate (OCR) on synaptosome

Purified synaptosomes were resuspended in 140 mM NaCl, 5 mM KCl, 5 mM $NaHCO_3$, 1 mM $MgCl_2$, 1,2 mM $Na_2HP0_4$, 10 mM D-glucose, 20 mM Hepes pH 7.4, and plated in XF24 cell culture microplates pre-coated with poly-D-lysine (100 µg/well). The plates were centrifuged at 3400 x g for 1 hour to ensure attachment of synaptosomes. Oxygen consumption was measured as for neurons (described in the main 'Material and Method' section), except that at the times indicated,

the injected compounds were: oligomycin (2 µM), fCCP (4 µM), Rotenone/Antimycin A (1 µM), and each measurement loop consisted of 30 sec mixing, 2 min incubation, and 2 min measurement of oxygen consumption.

## Glucose uptake

Glucose uptake measurements were performed on 15–17 DIV neurons, infected with shRNA against MPC1 or control shRNA as described above, and treated as indicated in the figure legends 30 minutes prior to performing the assay. Neurons were incubated with 2-NDBG (2-(N-(7-Nitrobenz-2-oxa-1,3-diazol-4-yl)Amino)–2-Deoxyglucose) in aCSF (described for the cell culture) containing 2 mM glucose for 15 min. After 5 min washing, live fluorescence (Ex 465 nm/ Em 540 nm) was quantified using Cytation three plate reader (BioTek Intruments).

## Immunostaining

Immunostaining was performed according to *De la Rossa et al., 2013* Briefly, neuro-MPC1-KO and neuro-MPC1-WT mice were perfused with 1 x PBS followed by 4% paraformaldehyde (PFA) in PBS and the brains where immediately post-fixed with 4% PFA at 4 °C overnight. Fixed brains were cut into 30 µm sections with a vibratome (Microm HM 650 V). The primary antibody was rabbit anti-MPC1 (Sigma, HPA045119, Anti-Brp44l); the secondary antibody was goat anti-rabbit Alexa Fluor (Life technologies, A11034) in presence of DAPI. Images were acquired using fluorescent (Zeiss Axiophot) or confocal (Zeiss LSM780) microscopy.

Determination of oxygen consumption rate (OCR) on synaptosomePurified synaptosomes were resuspended in 140 mM NaCl, 5 mM KCl, 5 mM NaHCO₃, 1 mM MgCl₂, 1,2 mM Na₂HP0₄, 10 mM D-glucose, 20 mM Hepes pH 7.4, and plated in XF24 cell culture microplates pre-coated with poly-D-lysine (100 µg/well). The plates were centrifuged at 3400 x g for 1 hour to ensure attachment of synaptosomes. Oxygen consumption was measured as for neurons (described in the main 'Material and Method' section), except that at the times indicated, the injected compounds were: oligomycin (2 µM), fCCP (4 µM), Rotenone/Antimycin A (1 µM), and each measurement loop consisted of 30 sec mixing, 2 min incubation, and 2 min measurement of oxygen consumption.

Immunostaining was performed according to *De la Rossa et al., 2013* Briefly, neuro-MPC1-KO and neuro-MPC1-WT mice were perfused with 1 x PBS followed by 4% paraformaldehyde (PFA) in PBS and the brains where immediately post-fixed with 4% PFA at 4 °C overnight. Fixed brains were cut into 30 µm sections with a vibratome (Microm HM 650 V). The primary antibody was rabbit anti-MPC1 (Sigma, HPA045119, Anti-Brp44l); the secondary antibody was goat anti-rabbit Alexa Fluor (Life technologies, A11034) in presence of DAPI. Images were acquired using fluorescent (Zeiss Axiophot) or confocal (Zeiss LSM780) microscopy.

The TUNEL assay was performed on fixed coronal sections using the DeadEnd Colorimetric TUNEL System (Promega, G7130) according to the manufacturer's instructions. For in vitro immunostaining, 15 DIV control and neurons infected for 7 days with MPC1-targeting shRNA or control control shRNA, were washed in HBSS and fixed in PBS containing 4% PFA and 4% sucrose for 15 min. Neurons were incubated in 5% goat preimmune serum (GPi) diluted in 50 mM Tris pH7.4, 0.2% Tween20, for 20 min, then incubated in anti-βIII-tubulin (Mouse, Biolegend-801201) with anti-MPC1 or TOMM20 (Rabbit, ab186735, Abcam) and antibodies for 1 h at room temperature. After several washes in PBS, the sections were further incubated in PBS, 5% GPi, 0.2% Tween20 containing goat anti-rabbit Alexa Fluor594 (Life technologies, A11037) and goat anti-mouse Alexa Fluor488 (Life technologies, A32723) in presence of DAPI. Finally, sections were air-dried and mounted as above and images were acquired by confocal microscopy (Zeiss LSM780).

## Western blotting

Cultured neurons, brain extracts or synaptosomes were homogenized in lysis buffer (50 mM Hepes, pH 7.4, 0.5% Triton X-100, protease inhibitors), and resolved by SDS-PAGE in 8% to 15% polyacrylamide gels. Proteins were electrotransferred to PVDF membranes. Membranes were blocked for 30 min in TBS containing 0.1% Tween20% and 5% milk powder. Primary antibodies were prepared in the same blocking solution and incubated with the membranes overnight at 4 °C. After three washes in TBS 0.1% Tween20, secondary HRP-coupled antibodies in blocking solution were added for 60 min with agitation at room temperature. After extensive washing, bound-antibodies were revealed using the ECL kit (Biorad). Antibodies used: anti-MPC1 (Rabbit, Sigma, HPA045119), anti-MPC2 (Mouse, Millipore, MABS1914), anti-MCU (Rabbit, Sigma, HPA 016480), anti-HSP70

(Mouse, Clone JG1, Millipore, MABS1955), anti-synaptophysin (Mouse, Abcam, ab8049), anti-Tyrosine Hydroxylase (Rabbit, Millipore, AB152), anti-CamKIIα (Goat, Abcam, ab87597), anti-GFAP (Mouse, Sigma, G3893), anti-VDAC (Goat, Santa Cruz, sc-8829), anti-Actin (Beta-actin-peroxidase, Sigma, a3854), anti-IgG-Rabbit-HRP (Dako, P0217), anti-IgG-Mouse-HRP Dako, P0447, anti-IgG-Goat-HRP (Santa Cruz, sc-2304).

