## [Editor Report]

This paper finds that conditional deletion in excitatory neurons of the only known pathway allowing the uptake of pyruvate into mitochondria, the mitochondrial pyruvate carrier, is tolerated in mice but produces hyperexcitability. The mice are more susceptible to epileptic seizures when inhibitory neurotransmission is impaired pharmacologically. Convincing evidence is presented that this hyperexcitability is due to decreased activity of Kv7.2/7.3 channels, secondary to dysregulation of cellular calcium handling.

---

## [Decision Letter]

[Editors' note: this paper was reviewed by Review Commons.]

---

## [Author Response]

Reviewer #1 (Evidence, reproducibility and clarity (Required)):The authors have presented a very interesting and compelling set of data regarding the impact of conditional deletion of the only known pathway allowing the uptake of pyruvate into mitochondria. The paper comprises two interwoven stories that are both important. The first is the remarkable finding that the majority of excitatory neurons in the cortex (i.e. those under the influence of the CaMKII promoter) show remarkable metabolic flexibility as they tolerate elimination of pyruvate oxidation, considered the major supplier of ATP in neurons. The data on this seem clear although the authors did not delve into the potential mechanisms of metabolic compensation that likely occurs. Instead they examined whether there was some mal-adaptive compensation and they found clear evidence of this: in the absence of MPC activity the mice are much more prone to epileptic seizures, unveiled experimentally by relatively standard protocols (kindling). The authors present largely very convincing evidence that this mal-adaptive compensation in turn ends up decreasing the activity of KV7.2/7.3 channels whose job is normally to limit runaway repetitive firing by mediating an hyperpolarizing K^+^ efflux following an action potential. This channel, put on the map as it was one of the downstream targets modulated by cholinergic metabotropic activation, is also know known to be controlled by Calmodulin and therefore cytosolic Ca levels. Overall, I think at its core this manuscript is interesting and important. There however several weaknesses, I fear, will diminish the impact on the eventual readership. If these points can be addressed, it will strengthen the longevity of these findings:1) It is puzzling why the authors resorted to using shRNA-mediated KD of MPC1 for some of the in vitro studies when they have gone to the trouble of making a floxed CRE-dependent mouse. Primary cells (e.g. Figure 1) or organotypic cultures (Figure 6) from these mice would have made a more consistent set of starting conditions to compare data across the manuscript. As there viruses expressing the CRE recombinase are widely available this could have been used on mice simply harboring the floxed gene it they are worried about waiting for the expression of the CaMKII promoter for in-vitro conditions.

This is indeed a good point. We initially tried to use viruses expressing the CRE recombinase in cultured neurons from mice harboring the floxed gene as proposed by the reviewer. However, for reasons that we do not understand, the use of AAVs or lentiviruses expressing the CRE was found to be deleterious for the cultured neurons. In view of this toxicity we tried using TAT-CRE recombinase, a recombinant cell-permeant fusion recombinase, which we added directly to the medium. However, this strategy proved to be poorly efficient. We finally used cultures of Cre-floxed neurons in which we tried to knockout MPC1 gene using 4-hydroxytamoxifen in the culture medium. However, we did not obtain satisfying results because, as previously reported, cortical neurons grow poorly in the presence of 4-hydroxytamoxifen (Nichols et al., Cell Death and Disease, 2018. https://doi.org/10.1038/s41419-018-0607-9). For all these reasons we turned to the shRNA strategy and to the use of 3 small molecule inhibitors of the MPC each with different chemical structures. Both the RNA interference and the pharmacological approaches gave similar results, reinforcing our confidence in the specificity of the results, and the unlikelihood of off-target effects.

2) The data in Figure 5 gets a little less convincing as using extracellular glutamate to drive Ca elevations is so non-physiological that the results might really be distorted by the participation of something irrelevant to the story, even though it supports the overall interpretation for a role of Ca/CaM in the control of the channel. Similarly, the use of RU360 should be done with caution. The drug, although a useful antagonist of MCU in purified mitochondria, is famously finicky with respect to its ability to cross membranes and could well have off target impact. A much cleaner experiment would be to suppress the expression of MCU via KD. Presumably in the MPC-deficient neurons, this would have minimal impact on Ca signals. Given the frequent ambiguity associated with interpreting pharmacological results, coupled to the central importance of this finding in interpreting the entire paper, I think carrying out experiments with molecular genetic manipulation of MCU is warranted.

We thank the Reviewer for this valuable comment. Following their recommendation, we genetically downregulated the MCU using RNA interference. As shown in Figure 6 – Supplementary Figure 2A in the revised manuscript, we used a shRNA against MCU (Qiu et al., Nature Communications, 2013, doi:10.1038/ncomms3034), and were able to significantly downregulate the expression of the MCU by 66% in cultured cortical neurons.

We then measured cytosolic calcium levels using the Fluo4-AM probe and mitochondrial calcium levels either directly using mitochondria targeted aequorin as recommended by the Reviewer (see minor comment 2) or indirectly, by monitoring cytosolic calcium upon mitochondrial depolarization with the protonophore fCCP. Upon fCCP-induced mitochondrial depolarization, mitochondria release their calcium content.

Using high KCl concentration (50 mM), we found that cytosolic calcium increased significantly upon depolarization of cultured cortical neurons. However, the signal was significantly higher in the presence of the MPC inhibitor Zaprinast compared to control. Similar to MPC inhibition, downregulation of the MCU expression resulted in increased cytosolic calcium levels compared to control neurons. Combination of MCU and MPC inhibition did not result in higher levels of cytosolic calcium levels compared to separate inhibition of these two transporters. When mitochondria were depolarized using fCCP, cytosolic calcium augmented dramatically in control neurons, suggesting that a high amount of calcium had been taken up by mitochondria upon KCl^-^induced neuron excitation. In contrast, the level of calcium released by mitochondria from neurons treated with Zaprinast and/or from MCU-deficient neurons was significantly reduced compared to control neurons, suggesting a lower capacity of mitochondria from these neurons to take up calcium (see Figure 6 – Supplementary Figure 2B-D).

To further assess the capacity of mitochondria from MPC and/or MCU-deficient neurons to take up calcium upon neuron activation, we measured the bioluminescence emitted by mitochondria targeted aequorin. We found decreased bioluminescence levels in MPC and/or MCU deficient neurons exposed to KCl, confirming that mitochondria from these neurons show a deficit in calcium mitochondrial intake (see Figure 6G in the revised manuscript). These results likely explain that MPC-deficient neurons show abnormal elevated calcium levels in the cytosol upon stimulation.

These new results consolidate our previous findings using RU360 and support our hypothesis that the lower mitochondrial membrane potential of MPC-deficient neurons, as shown in our paper, is responsible for low calcium import by the MCU upon neuronal activation. This would explain, at least in part, the higher cytosolic calcium level in depolarized MPC- deficient neurons. High cytosolic calcium would reduce M-type channel activity, as previously published, and would explain the hyperactivation of neurons upon intense firing.

3) The authors have not really made clear in this paper whether the ability to suppress the phenotype of the MPC deficiency with ketones is really related to a providing TCA cycle support or instead a pharmacological impact on non-TCA related targets (such as the Kv7.2/7.3 channels). Presumably the use of other ketones might circumvent this. The action of ketone bodies has been a topic of considerable interest in neuroscience, given the clinical relevance for childhood epilepsies. Previous studies for example have argued for direct inhibition of the vesicular glutamate transporter (Juge et al. Neuron 2010). The use of other ketones (acetoacetate) would narrow down the interpretations of the data.

Our results point to 2 two possible mechanisms of 3-beta hydroxybutyrate: (i) providing acetyl-CoA to the Krebs cycle, thereby stimulating OXPHOS and (ii) direct action of 3-beta hydroxybutyrate on the activity of Kv7/7.3 channels. The reviewer is asking whether, in addition to 3-beta hydroxybutyrate, other ketone bodies, acetone or acetoacetate, may display antiepileptic activity, which would probably indicate that providing substrates to the TCA cycle is sufficient to prevent neuron-intrinsic hyperactivity and seizures. This is an interesting question, which we have now addressed.

We found that acetoacetate, 1% supplemented in the drinking water for one week before PTZ administration, was able to prevent the occurrence of seizures induced by PTZ (see Figure below). This result indicates that providing acetyl-CoA to mitochondria to fuel the TCA cycle and boost OXPHOS is sufficient to prevent the occurrence of seizures after PTZ administration. We thank the Reviewer for suggesting us this important experiment, which we have now added in the manuscript (Figure 4D).

Other:1) In vitro – scramble controls only serve to demonstrate there is no general effect of treating cells with shRNAs, but do not address if there is an off-target effect. The most convincing thing here would be to have an shRNA-insensitive variant that rescues.

We have used 2 different shRNAs and 3 chemically unrelated inhibitors of the MPC and in all cases we obtained similar results. Therefore, we think that it is unlikely that the effects we observe are due to an off-target activity. The experiment proposed by the reviewer is interesting but extremely difficult. The idea would be to reintroduce a shRNA-insensitive MPC1 into MPC1-deficient neurons treated with shRNA. This is difficult as it is known that the expression level of MPC1 needs to be matched to that of MPC2, otherwise it leads to depolarization of the mitochondria. Obtaining the right level of MPC1 would be extremely difficult to achieve in practice.

2) Does rescuing CaMK binding to KCNQ channels rescue the phenotypes?

The question raised by the Reviewer implies that CaM is not constitutively bound to KCNQ channels, which is a matter of debate. As we pointed out in the discussion, ‘Intracellular calcium decreases CaM-mediated KCNQ channel activity (32, 36) by detaching CaM from the channel or by inducing changes in configuration of the calmodulin-KCNQ channel complex (36).’ The CaM-KCNQ tethering is also described in a review by Alaimo and Villaroel, 2018 (doi:10.3390/biom80300579): ‘[…] CaM was first defined as an integral subunit constitutively tethered to the C-terminal region of Kv7.2/3 channels since Kv7.2 mutants that were deficient in CaM binding were unable to generate measurable currents [5,21]. However, this model has been questioned since Kv7.2 channels, carrying a hB mutation [40] or Kv7.4 hA mutated channels [41] that do not bind CaM, can still reach the plasma membrane and are functional.’

When considering to manipulate CaM binding to KCNQ, it should also be considered that previous studies on this matter have been performed with heterologous systems and through genetic manipulations of CaM (by expression of a dominant negative or by overexpression of CaM) or of the KCNQ binding motif.

Based on both theoretical and practical issues, we, thus, believe that it is not feasible to implement a straightforward approach that would be compatible with our mouse model.

An alternative to this experiment was proposed by Reviewer #3, who recommended to use Ca^2+^ chelators in our electrophysiological experiments. This is the option we have chosen and we kindly ask the Reviewer to refer to point 1d raised by Reviewer 3.

3) As the authors imply that BHB activates KCNQ channels, showing this directly in their prep would provide some convincing data. If this is true, why doesn’t BHB increase firing rate of WT neurons?

Activation of KCNQ channels is expected to reduce (not increase) neuronal firing. When exposed to BHB, we indeed found that WT cells also show a trend towards decreased excitability (p=0.08). We report this trend now in the revised figure 5F. Given that KCNQ channels are already available to be recruited upon repetitive firing in WT cells (to a larger extent as compared to KO, as indicated by our data with XE991) it is conceivable that a further potentiating effect of BHB at the concentration used for ex vivo recordings (2 mM) will be limited.

4) How does the anti-epileptic effects of ketones in this study relate to previous reports of regulation of KATP channels? One of main concerns is that ketones might have a parallel anti-epileptic effect in the MPC1 KO mice that is unrelated to the mechanism proposed here.

The mechanisms by which ketones decrease neuron firing has been debated for several years and several mechanisms have been proposed, including inhibition of glycolysis and activation of ATP-sensitive potassium channels (KATP channels) as pointed out by the reviewer. We do not exclude at all the possibility that inhibition of the MPC could also have an impact on the KATP channels, especially because we observed that inhibition of the MPC activity leads to increased aerobic glycolysis, a process that is prevented by 3-beta hydroxybutyrate (Figure 1 of our paper). We have been investigating for quite a long time whether KATP channel activity was downregulated in MPC-deficient neurons and whether we could detect increased ATP near the plasma membrane. So far, we have not obtained convincing data showing a reduced activity of KATP channels in MPC-deficient neurons. These results however do not exclude the possibility that KATP channels, in addition to the M-channels, are involved in the hyperexcitability of MPC-deficient neurons and could be direct or indirect targets of ketones. This is now mentioned in the Discussion (line 460-478 p20):

“The ketogenic diet has been reported to decrease seizures in patients with pharmacologically refractory epilepsy (Carroll et al., 2019) and we now report that the hyperexcitability of neuro-MPC1-KO mice fed with ketones is significantly reduced. […] Whether KATP channels are involved in the protection conferred by ketones in neuro-MPC1-KO mice requires further investigations.”

Minor comments:1. What is the MPC1 KO efficiency in CaMK neurons? The western blot in 2c is from the whole cortex and therefore does not show that.

As shown in figure 2C, MPC1 is significantly decreased in synaptosomes isolated from MPC1 KO cortices. This figure shows that these synaptosomes are highly enriched for CamKII and contain less astrocytic marker GFAP in comparison to the whole cortex.

2. Mitochondrial Ca^2+^ levels are not measured directly, for which there are many tools. This is needed to demonstrate definitively that there is a defect in Ca^2+^ handling."

The reviewer raises an important point and as mentioned above we have now used the mito-Aequorin, a luminescent quantitative probe targeted to mitochondria (M. Bonora et al., Subcellular calcium measurements in mammalian cells using jellyfish photoprotein aequorin-based probes. Nat Protoc 8, 2105-2118 (2013); A. Tosatto et al., The mitochondrial calcium uniporter regulates breast cancer progression via HIF-1alpha. EMBO Mol Med 8, 569-585 (2016)).

This is described above, in the response to Major point 2.

Reviewer #2 (Evidence, reproducibility and clarity (Required)):De la Rossa and colleagues examined the consequences of conditionally knocking out MPC1,a subunit of the mitochondrial pyruvate carrier. They found that despite decreased levels of oxidative phosphorylation in excitatory neurons, phenotypically these conditional knockout mice were normal at rest. However, when challenged by inhibition of GABA neurotransmission, these animals developed severe seizure activity and expired. These authors then showed that neurons with an absence of MPC1 were hyperexcitable in part through abnormal calcium homeostasis, which was associated with a reduction in M-type inhibitory potassium channel activity. Intriguingly, the ketogenic diet and the major ketone body beta-hydroxybutyrate were able to reverse these changes.This is a carefully conducted research study that reveals cell type-specific alterations of MPC1 deletion and functional consequences. The study design was logical and involved an exhaustive array of methodologies. The manuscript was generally well written and organized, and there are no major concerns. This study shows a direct causal relationship between impaired bioenergetics at the level of mitochondrial, and subsequent behavioral seizures, and is perhaps the most direct demonstration to date that an intrinsic disturbance of metabolic function can result in seizure activity (through changes in calcium regulation and impairment of ion channel activity). This will be an important contribution to the scientific literature.Minor comments:1. Page 4, line 86: Would recommend changing "paroxystic" to "paroxysmal" (the latter which is a more recognized term).

The change has been made and can be found in page 3 line 74 of the revised manuscript.

2. Page 5, line 124: recommend including the concentration of beta-hydroxybutyrate used when first mentioned. In general, concentration and dose information were difficult to find, as well as route of administration (for kainate, page 7, line 175). This type of information was not conveniently presented.

We have added this information, as recommended by the Reviewer. See line 169 (p7) for Kainic acid and line 118 (p5) for BHB.

3. Page 5, line 128: “both overcomed” is awkward. Would recommend using “both reversed”.

The change has been made. See line 123 (p6) of the revised manuscript.

4. Page 8, line 193: the authors probably meant "astro-MPC1-WT mice", not "neuro-MPC1-WT mice".

The controls in these experiments were neuro-MPC1-WT mice.

5. Page 12, lines 280-282: the authors might want to mention that chronic exposure of BHB might reduce the hyperexcitability of neuro-MPC1-KO mice.

We have hesitated in introducing this comment in the revised manuscript given that the mice were fed a maximum of 7 days with a ketogenic diet and we were not sure that this could be considered a chronic exposure.

6. Please review entire manuscript and use consistent tense. For example, on page 13, line 309, to maintain the past tense, it should read “We first assessed whether.”"

We checked the entire text and tried to use consistent tense.

7. Page 13, line 318: the authors used 10 mM BHB when examining calcium levels, but they earlier used 2 mM. They need to explain why they used a different concentration; and 2 vs 10 mM are quite different.

The reviewer makes a valid point. When we performed the in vitro experiments, we used 10 mM BHB, which is slightly higher than the amount of ketone bodies measured in the blood of mice fed on a ketogenic diet for 2 days (Supplementary figure 4). This concentration of BHB has also been used in other studies (see for example: Izumi et al., JCI 1998, 101:1121-1132). Later on, when electrophysiology experiments were performed, the person in charge of these experiments followed a previously published protocol by Yellen and colleagues, in which the authors had used 2 mM BHB (Ma et al., J. Neurosci 2007,27: 3618-3625). This explains the differences between the concentrations used in vitro and in vivo. The reference of Ma et al. was added lines 238-239, page 10 of the revised manuscript.

8. Page 13, line 323: it is not necessary to say “……interesting study published during the preparation of this manuscript." This phrase should be deleted, and the relevant reference simply cited.

We now removed this sentence

9. The authors need to explain more clearly in the beginning what exactly is meant by "paradoxical" hyperactivity. They provide greater meaning later in the manuscript, but this should be clarified at the outset.

We have explained what we mean by ‘paradoxical’ in the Introduction, line 72 (p3), page 3, and removed paradoxical from the abstract.

Reviewer #3 (Evidence, reproducibility and clarity (Required)):2(MPC) in regulation of neuronal excitability. The authors find that MPC deficiency in glutamatergic neurons is associated with aerobic glycolysis, inhibition of the M-type K channels, and neuronal hyperexcitability that manifests in increased sensitivity to chemical pro-convulsants without changes in resting conditions. Alterations in Ca homeostasis in MPC-deficient neurons is consistent with reduced mitochondrial membrane potential and attendant diminution of mitochondrial calcium buffering capacity. The authors further show that the effect of MPC deficiency can be phenocopied by treatment of wild type neurons with a chemical inhibitor of the mitochondrial Ca uniporter (MCU). Based on these data, it is proposed that reduced mitochondrial Ca uptake causes neuronal hyperexcitability in the absence of MPC. Overall, the manuscript presents detailed electrophysiology and in vivo seizure studies. However, there is significant disconnect between the actual data in Figure 6 and the authors’ conclusions/proposed mechanism. In particular, the evidence for the role of Ca in the hyperexcitability due to MPC deficiency is the weak link in the authors’ argument.1. The studies linking reduced mitochondrial Ca uptake to hyperexcitability in MPC-deficient neurons (Figure 6) have several limitations that significantly weaken the paper:1a. The Ca measurements in cortical neurons (Figure 6A-F) are performed under conditions (glutamate/KCl) that are fundamentally different from those used in electrophysiology of CA1 pyramidal neurons (Figure 6G-N). The electrophysiological excitation is much briefer and less extreme than the chemical stimulation, and it is not clear that the Ca dysregulation occurs at the earliest times (see Figure 6A).

The use of glutamate/KCl to depolarize cultured neurons has been very useful to unmask a defect in calcium import into MPC-deficient mitochondria. This point is crucial and has now been consolidated with new experiments performed using downregulation of the MCU and the use of mitochondria targeted aequorin. However, we agree that this does not answer the key question raised by the Reviewer: does calcium dysregulation occur at the earliest time and does this play a role in neuron hyperexcitability? In Comment 1d, the Reviewer asked to question the role of calcium using a calcium chelator in the electrophysiological experiments. This is an important comment, which we have now addressed as detailed below under point 1d. These new results clearly establish an importance of calcium as a key factor in the hyperexcitability of MPC-deficient neurons and strongly suggest that calcium dysregulation initiates this process.

1b. The conclusion that MCU is functionally responsible for MPC’s effect on neuronal excitability is singularly based on the use of RU360 as a chemical inhibitor of MCU but the specificity of this reagent is questionable. Evidence for a cause and effect relationship that directly implicates altered MCU/mitochondrial Ca buffering has not been provided.

We thank the Reviewer for this valuable comment, which was also mentioned by Reviewer1. Following their recommendation, we genetically downregulated the MCU using RNA interference. As shown in Figure 6 – Supplementary Figure 2A of the revised manuscript, we used a shRNA against MCU (Qiu et al., Nature Communications, 2013, doi:10.1038/ncomms3034), and were able to significantly downregulate the expression of the MCU by 66% in cultured cortical neurons.

We then measured cytosolic calcium levels using the Fluo4-AM probe and mitochondrial calcium levels either directly using mitochondria targeted aequorin as recommended by the Reviewer (see minor comment 2) or indirectly, by monitoring cytosolic calcium upon mitochondrial depolarization with the protonophore fCCP. Upon fCCP-induced mitochondrial depolarization, mitochondria release their calcium content.

Using high KCl concentration (50 mM), we found that cytosolic calcium increased significantly upon depolarization of cultured cortical neurons. However, the signal was significantly higher in the presence of the MPC inhibitor Zaprinast compared to control. Similar to MPC inhibition, downregulation of the MCU expression resulted in increased cytosolic calcium levels compared to control neurons. Combination of MCU and MPC inhibition did not result in higher levels of cytosolic calcium levels compared to separate inhibition of these two transporters. When mitochondria were depolarized using fCCP, cytosolic calcium augmented dramatically in control neurons, suggesting that a high amount of calcium had been taken up by mitochondria upon KCl^-^induced neuron excitation. In contrast, the level of calcium released by mitochondria from neurons treated with Zaprinast and/or from MCU-deficient neurons was significantly reduced compared to control neurons, suggesting a lower capacity of mitochondria from these neurons to take up calcium (see Figure 6 – Supplementary Figure 2B-D).

To further assess the capacity of mitochondria from MPC and/or MCU-deficient neurons to take up calcium upon neuron activation, we measured the bioluminescence emitted by mitochondria targeted aequorin. We found decreased bioluminescence levels in MPC and/or MCU deficient neurons exposed to KCl, confirming that mitochondria from these neurons show a deficit in calcium mitochondrial intake (see Figure 6G in the revised manuscript). These results likely explain that MPC-deficient neurons show abnormal elevated calcium levels in the cytosol upon stimulation.

These new results consolidate our previous findings using RU360 and support our hypothesis that the lower mitochondrial membrane potential of MPC-deficient neurons, as shown in our paper, is responsible for low calcium import by the MCU upon neuronal activation. This would explain, at least in part, the higher cytosolic calcium level in depolarized MPC- deficient neurons. High cytosolic calcium would reduce M-type channel activity, as previously published, and would explain the hyperactivation of neurons upon intense firing.

1c. There is a large variation in the effect of 10 μm RU360 on firing frequency, comparing Figure 6H and N (blue traces), including the shape of the traces and values at ramp number 6. This calls into question the reliability of the comparisons in each separate figure.

**Author response image 1. sa2fig1:** 

Data presented in each single graph in the main Figures were obtained from groups of littermates through recordings conducted in consecutive days. Some caution is warranted when comparing data between different figures (i.e. between different experimental series), as several factors may contribute to inter-experiment variability, including variability between different batches of animals. However, the difference pointed out by the reviewer regarding the values of cell firing reported in Figure 6I and O is only apparent. When applying depolarizations with ramps of 5s, a fair amount of WT cells infused with RU-360 show high instantaneous firing frequency, especially for the last ramps that steeply reach high current levels. This leads to accommodation/inactivation of the action potential towards the end of the ramps, as shown in the example trace in Figure 6H. As a result, the current-frequency plot deviates from linearity, as it is the case in Figure 6I (blue trace) and, even more evidently, in Figure 6O. We have now reanalyzed the same recordings from WT cells infused with 10 µM RU-360 and measured the firing frequency in response to a square depolarizing step (250 pA) of 0.5 or 1 second. No difference was found between the firing frequencies of the cells from Figure 6I and Figure 6O (group 1 and group 2, respectively, in Author response image 1). Although the ramps may lead to some distortion for higher stimulation levels, we have decided to show results from ramps consistently throughout the main figures because this protocol with continuously increasing currents allows us to measure more precisely the rheobase and the firing threshold (as opposed to the stepwise increments of a square stimulation).

1d. The calcium > PIP2 > M-type K^+^ channel axis is well established but has not been fully explored in the context of MPC deficiency. The use of a calcium chelator will likely be informative in this context, and would be better evidence for a role of Ca in the MPC effects.

We would like to thank the Reviewer for this useful comment and suggestion. As recommended, we used a calcium chelator to further substantiate the hypothesis that higher cytosolic calcium impairs M-type K^+^ channel activity in neuro-MPC1-KO cells. We performed patch-clamp experiments using an intracellular solution containing the calcium chelator BAPTA (10 mM). We compared cell firing between the first minutes after the establishment of the whole-cell condition and 15 minutes thereafter, i.e. before and after complete diffusion of BAPTA in the cytoplasm. Whereas intrinsic excitability was unaltered in neuro-MPC1-WT cells at these two timepoints, neuro-MPC1-KO cells displayed significantly lower firing rate and higher rheobase upon calcium chelation (Figure 6P-R of the revised manuscript). Notably, unlike in BAPTA-free recordings (Figure 5J-L), subsequent bath application of XE991 (10 µM) increased intrinsic excitability not only in neuro-MPC1-WT cells, but also in neuro-MPC1-KO cells, indicating that calcium chelation made available a contribution of M-type K^+^ channels that was otherwise impaired by the high intracellular calcium levels (cf. Figure 5J-L).

1e. The ability of BHB to rescue various parameters in this and other figures in the paper is interesting but does not directly speak to the specific mechanism as to how MPC deficiency affects neuronal excitability. BHB’s effect is consistent with the metabolic flexibility of neurons when the TCA cycle cannot be fueled by glucose/pyruvate (as in GLUT1 or MPC deficiency).

In response to comment 3 of Reviewer 1, we have tested whether another ketone body, acetoacetate, could prevent PTZ-induced seizures in MPC-deficient animals. We found that this is the case (see Figure 4D in the revised manuscript). Given that this ketone body is not expected to directly bind the M channel, the most likely hypothesis as to how ketone bodies prevent PTZ-induced seizures in MPC KO neurons is through fueling the TCA cycle with acetyl-CoA and boosting OXPHOS.

Our hypotheses regarding the mechanism by which MPC deficiency affects neuronal excitability and ketones prevent this is as follows:

The MPC allows mitochondria to use pyruvate as a substrate for respiration and ATP production through OXPHOS. Its absence leads to decreased oxygen consumption and ATP production as shown by us and Divakaruni et al. (JCB, 2017). Furthermore, we show here that mitochondria display a lower mitochondrial membrane potential and a decreased capacity to import calcium. Upon intense firing, cytosolic calcium would increase, impairing the activity of M channels, and resulting in increased excitability. We show that ketones boost OXPHOS, prevent increased cytosolic calcium and thereby maintain M channels functional.

This mechanism does not exclude other mechanisms as it is likely that the effects of ketones are probably multifunctional. In particular, given that we observed increased glucose uptake in MPC-deficient neurons (Figure 1) and that ketones prevent this (Figure 1), as other groups before us (i.e. the labs of Garry Yellen and of Nika Danial) have shown, we have thought about the involvement of KATP channels. We have been investigating for quite a long time whether KATP channel activity was downregulated in MPC-deficient neurons and whether we could detect increased ATP near the plasma membrane. So far, we have not been able to convincingly show a reduced activity of KATP channels in MPC-deficient neurons. These results however do not exclude the possibility that KATP channels, in addition to the M-channels, are involved in the hyperexcitability of MPC-deficient neurons and could be direct or indirect targets of ketones. This is now mentioned in the Discussion and reads as follows (line 460-478 p20):

“The ketogenic diet has been reported to decrease seizures in patients with pharmacologically refractory epilepsy (Carroll et al., 2019) and we now report that the hyperexcitability of neuro-MPC1-KO mice fed with ketones is significantly reduced. KATP[…] Whether KATP channels are involved in the protection conferred by ketones in neuro-MPC1-KO mice requires further investigations.”

2. The manuscript (and the field) will benefit from a more scholarly discussion and integration of published literature:

We have modified the Discussion. All changes are in red and added some essential references.

2a. The published studies on the outcome of pharmacologic MPC inhibition in neurons (Ref 18, Divakaruni et al.) are not only consistent with the bioenergetic effect in Figure 1, but more importantly, show that interference with MPC does not lead to broad deficiencies in energy metabolism but rather remodel fuel utilization patterns to alternative substrates that feed the TCA cycle (BHB, leucine, etc). For this reason, terms such as “mitochondrial dysfunction” and “OXPHOS deficiency” used throughout the manuscript to describe MPC deficiency are vague and imprecise. In addition, this metabolic flexibility may explain lack of defects under resting conditions. In light of these considerations, the argument as to whether aerobic glycolysis in MPC-deficient neurons explains the lack of phenotype in resting conditions (p 17) seems one-sided. Overall, the studies in ref 18 are relevant to the current manuscript and should be better integrated in the discussion.

We agree with the Reviewer that the work of Divakaruni et al. provides an explanation for how, using glutamine oxidation, and possibly the branched chain keto-acid catabolites of leucine, isoleucine, and valine, neurons could compensate for mitochondrial pyruvate impairment, at least under resting conditions. These results are now mentioned in the Discussion as recommended by the Reviewer (see lines 396-413 p17-18) and we thank them for suggesting this very interesting hypothesis.

Regarding OXPHOS deficiency, as mentioned in the Divakaruni et al. paper, neurons lacking MPC activity show decreased maximal respiratory capacity and a drop in ATP levels, which, indeed, is consistent with our results (Figure 1). This is what we refer to OXPHOS deficiency throughout the text. However, to be more precise, when possible we replaced OXPHOS deficiency by drop in oxygen consumption and ATP production. So we agree that mitochondria with MPC impairment could, through metabolic flexibility, maintain a normal function under resting conditions, but on the other hand, when neurons are firing intensively, as is the case in the presence of PTZ, we think that mitochondria do not cope properly with the energetic demand and this, in part, results in a deficit in calcium intake.

2b. Several references are cited to describe the role of OXPHOS vis-à-vis aerobic glycolysis in neuronal function. At times, however, the authors’ statements are not consistent with what these papers actually show (or do not show). For example, see the use of refs 6 and 44 on p17 of the discussion, where the authors state that aerobic glycolysis uncoupled from OXPHOS is sufficient to provide ATP for normal neurotransmission, but this does not mean OXPHOS is not needed.

We thank the Reviewer for noting these inconsistencies. We have now removed these inappropriate references from the manuscript.

2c. Although the XE991 experiments support an important role for the M-type channels in the altered excitability with deficiency, it is not clear that the proposed mechanism can explain all of the electrophysiological differences, particularly those resting properties that are measured without a Ca challenge to the neurons. It would be good to discuss other possible mechanisms that could affect neuronal excitability.

Our results point to M-type channels as important players in the phenotype of the MPC-deficient mice. Previous reports indicate that inhibition of this channel by XE991 can modulate input resistance, membrane potential and firing threshold of pyramidal cells (e.g. Shah et al., 2018, doi/10.1073/pnas.0802805105; Hu et al. 2007, DOI:10.1523/JNEUROSCI.4463-06.2007; Petrovic et al., 2012, doi:10.1371/journal.pone.0030402). We also found that XE991 induced a shift towards more negative potentials in the firing threshold of WT cells, but not in MPC1 deficient cells (-3.3±0.6 vs. -0.4±1.0, n=9, 8, p=0.027). However, we agree with the reviewer that the phenotype is probably highly complex and that additional mechanisms may contribute to modulate the intrinsic excitability of MPC-deficient neurons. As recommended by the Reviewer, and as already mentioned above, additional mechanisms have been discussed in the Discussion from the revised manuscript.